# Integrated RNA-seq and scRNA-seq to explore the biological mechanisms of mitophagy-related genes in ulcerative colitis

**Jianguo Ma[1]☯, Rongyi Xu[2]☯, Yunfei Liu[3], Junli Shao[2], Jing Xu[2], Yan Qi[2]\***

**1** Department of Proctology, The First Affiliated Hospital of Yunnan University of Chinese Medicine/ Yunnan Provincial Hospital of Traditional Chinese Medicine, Kunming, China, **2** Central Laboratory, The First Affiliated Hospital of Yunnan University of Chinese Medicine/Yunnan Provincial Hospital of Traditional Chinese Medicine, Kunming, China, **3** Department of Gastroenterology, The First Affiliated Hospital of Yunnan University of Chinese Medicine/Yunnan Provincial Hospital of Traditional Chinese Medicine, Kunming, China

☯ These authors contributed equally to this work and share first authorship.
\* qiyankm@163.com

## Abstract

Mitophagy's role in ulcerative colitis (UC) is not fully understood. This study explores mitophagy's impact on UC and aims to create a diagnostic model. The transcriptomic datasets of patients with UC and healthy controls were obtained from the Gene Expression Omnibus database. The mitophagy-related differentially expressed genes (MRDEGs) and hub genes were identified by weighted gene co-expression network and protein-protein interaction (PPI) network analysis. Networks of mRNA-miRNA and mRNA-TF were established to detect pertinent mechanisms. CIBERSORT was used to evaluate association between immune cells with hub genes. A diagnostic model was developed utilizing logistic regression. Single-cell RNA sequencing was used to characterize the expression of hub genes in specific cell clusters and the results of the differential analysis were annotated with the hub genes identified in the logistic model. Finally, a mice model of colitis was established, and the results were verified using qRT-PCR and western blot. The study identified 28 MRDEGs and 20 hub genes. A significant link was found between immune cell infiltration and hub genes, highlighting mitophagy's interaction with the immune response. A diagnostic model with 13 potential markers was developed, achieving high accuracy. Single-cell RNA sequencing delineated key cell types and confirmed varied hub gene expression, with validation through RT-qPCR and western blot. These findings not only deepens our understanding of mitochondrial autophagy in UC but also establishe a robust diagnostic model through interdisciplinary approaches, laying the groundwork for the development of targeted diagnostic and therapeutic strategies that warrant further research for clinical application.

**Data availability statement:** The publicly available datasets (GSE36807, GSE38713, GSE47908 and GSE231993) used for this study can be found in the GEO database.

**Funding:** The National Natural Science Foundation of China (Grant number [81960868]) and the Yunnan Provincial Science and Technology Department, funded by the Applied Basic Research Joint Special Funds of Yunnan University of Chinese Medicine (Grant numbers [202001AZ070001-051] and [202101AZ070001-013]), High-level TCM talents: (reserve talents) project of Yunnan (2021. No. 1), Xingdian Talent Support Program-Youth Talent Special Project (2023. No. 166), Yunnan Provincial Science and Technology Department - Yunnan University of Traditional Chinese Medicine Joint special Project (Grant numbers [202201AG070192]), Applied Basic Research Key Project of Yunnan (202501AS070155), Yunnan University of Chinese Medicine Joint Fund Project(XYLH2025001).

**Competing interests:** The authors have declared that no competing interests exist.

**Abbreviations:** UC, ulcerative colitis; MRGs, mitophagy-related genes; TF, transcription factor; PCA, principal component analysis; WGCNA, weighted gene coexpression network analysis; MF, molecular function; LASSO, least absolute shrinkage and selection operator; AUC, area under the curve; UMAP, uniform manifold approximation and projection; HSC, hemopoietic stem cell.

## Introduction

Ulcerative colitis (UC) is a progressive condition that can lead to various gastrointestinal issues, especially colon cancer, and can deteriorate the quality of life of individuals [1]. Nonetheless, the exact etiology of UC remains incompletely understood, although it might be associated with immune responses, gut bacterial imbalance, genetic susceptibility, and lifestyle factors [2]. Medical and surgical interventions increase the likelihood of relapse or adverse effects [3]. Hence, it is imperative to extensively comprehend the etiology of UC and detect possible molecular biomarkers to improve patient prognosis, diagnosis, and treatment strategies, decreasing the likelihood of disease relapse and complications.

Mitophagy is a form of selective autophagy that clears damaged or excessive mitochondria through the lysosomal pathway to maintain mitochondrial quality control and cellular homeostasis. This process plays a critical role in energy metabolism, oxidative stress regulation, and apoptosis, and is closely associated with various inflammatory and metabolic diseases [4]. Moderate mitophagy helps eliminate damaged mitochondria and limit the accumulation of reactive oxygen species (ROS), thereby reducing cellular inflammatory responses and maintaining epithelial barrier function [5]. However, when the mitophagy process is impaired or overactivated, it may lead to energy metabolism disorders, cellular dysfunction, and exacerbated tissue inflammation. Recent studies have revealed that mitochondrial dysfunction and abnormal mitophagy play significant roles in the pathogenesis and progression of inflammatory bowel disease (IBD), particularly in ulcerative colitis (UC), where it manifests as a coexistence of mitochondrial damage and immune dysregulation in intestinal epithelial cells. Multiple molecular pathways (PINK1/Parkin, BNIP3/NIX, FUNDC1) have been demonstrated to participate in the regulation of IBD-related mitophagy [6], suggesting a dual effect of this process in intestinal inflammation: it may exert a protective role by removing dysfunctional mitochondria, or it may exacerbate inflammatory responses due to regulatory imbalances. However, direct evidence from patient-derived tissue samples remains limited, and the diagnostic and therapeutic potential of mitophagy-related genes in UC requires further investigation.

Various bioinformatic techniques were employed in this study to identify and assess potential central candidates among the genes related to mitochondrial autophagy. The mitophagy-related gene (MRG) score of each disease group in the UC dataset was obtained using the ssGSEA algorithm. WGCNA was used to analyse the UC dataset, and hub genes were chosen from six related significant modules. These hub genes were then intersected with mitophagy-related differentially expressed genes (MRDEGs) and included in subsequent analyses. CIBERSORTx immune cycle analysis was conducted, and a diagnostic LASSO model was built for the UC dataset using the hub genes. The hub genes selected using the LASSO model were screened using a univariate logistic model. Additionally, the single-cell deletion dataset GSE231993 was used to perform a differential analysis of cell clusters between the control and UC groups, and the hub genes in the diagnostic model were tagged. Finally, we conducted gene expression analysis of 13 key genes from the diagnostic model in both UC mice and normal mice. The findings of this study will

aid in the identification of novel diagnostic biomarkers and therapeutic targets for UC, improving our understanding of its pathogenesis.

## Materials and methods

### Data download

This study included three colonic tissue transcriptome microarray cohorts (GSE36807, GSE38713, GSE47908) and one single-cell transcriptome sequencing cohort (GSE231993), all sourced from the Gene Expression Omnibus (GEO) public database. The experimental types for GSE36807, GSE38713, and GSE47908 were 'array-based expression profiling', utilizing the Affymetrix Human Genome U133 Plus 2.0 Array (GPL570) platform to provide whole transcriptome expression profiles of ulcerative colitis and control colonic mucosal samples. The experimental type for GSE231993 was 'high-throughput sequencing-based expression profiling', based on the Illumina NextSeq 500 platform (GPL18573), which yielded single-cell RNA sequencing data for colonic tissues. This study included all samples that met the criteria. The probe name annotations in the dataset were derived from the GPL platform files of the corresponding microarrays. Relevant information about the dataset is shown in S1 Table.

Final analysis sample (gene) exclusion criteria:

1. Genes with inconsistent symbols or those failing to match the HGNC nomenclature standards;

2. Genes with missing values or duplicate entries across data sources;

3. Genes with P-value ≥0.05 and no significant expression changes (logFC = 0) in differential expression analysis;

4. Non-protein-coding genes related to mitochondrial autophagy;

5. Genes without documented evidence of association with BPD in literature or databases.

Using the GeneCards database (https://www.genecards.org/), we queried genes associated with epigenetics using 'mitophagy' as the keyword. After applying a relevance score threshold of > 1, we identified 772 genes related to mitophagy. Additionally, we searched the Molecular Signatures Database(https://www.gsea-msigdb.org/gsea/msigdb/) for gene sets related to mitophagy in the Homo sapiens 'all collections' using the same keyword and identified four such gene sets. In total, 34 genes confirmed to be associated with mitophagy were identified. This study analysed 772 mitophagy-related genes (MRGs) sourced from these two databases, with the specific gene names listed in Supplementary S2 Table.

### Differential expression analysis

To determine the possible ways in which DEGs function in UC and their associated biological traits and pathways, we initially standardized the GSE36807, GSE38713, and GSE47908 datasets using the limma R package. Subsequently, we eliminated the batch effect of the UC datasets GSE36807, GSE38713, and GSE47908 using the R package sva. Graphs illustrating the contrast between the datasets before and after batch elimination using principal component analysis. To obtain the differentially expressed genes (DEGs) among the different groups in the UC dataset, we divided them into a disease group (UC) and a normal group (control group). Upregulated genes were those with logFC > 0.5 and P < 0.05, while downregulated genes were those with logFC < −0.5 and P < 0.05. The volcano diagram created using the R package ggplot2 shows the results of the variance analysis. After obtaining the DEGs, we compared them with the MRGs and created a Venn diagram to visualize the intersection. Next, we created a comparative diagram for the UC dataset, highlighting the genes that exhibited significant differences for further analysis. Additionally, we used the R package pheatmap to generate a heatmap to visualize gene expression.

## Constructing MRG scores for the ssGSEA algorithm

The ssGSEA algorithm measures the proportional representation of individual genes within a dataset. The ssGSEA algorithm was employed to assess the MRGs of each sample in the UC dataset using the R package GSVA. The score can be used to categorize samples from the disease group into high and low scores, enabling correlation analysis.

## Analysis of gene coexpression networks with weighted gene coexpression network analysis (WGCNA)

Initially, we computed the correlation coefficient among every pair of genes and utilized the weighted correlation coefficient value to ensure that the connections between genes in the network followed a scale-free network distribution. Gene correlation coefficients were used to construct hierarchical clustering trees. Distinct sections of the cluster tree symbolize various gene modules, substituted with diverse colours, followed by the computation of the module significance. For this study, the genes with the highest 25% gene expression variation in the UC dataset were utilized as inputs. The minimum requirement for the module genes was set at 100, whereas the ideal power threshold (soft power) was set at 20. The module-merging shear height was set to zero, indicating no module merging, and the minimum distance was set at 0.2. Subsequently, we computed the correlation between the MRG scores of patients with UC in the UC dataset and various modules. We then documented the genes present in each module, which were considered the characteristic genes of the module. After determining the modules with significant correlations ($P < 0.05$, $r \geq 0.30$) based on the correlation coefficient values, we intersected them with the MRDEGs. This intersection provided hub genes of the mitochondrial autophagy module, which were selected for further analysis.

## PPI interaction network

For this investigation, we constructed a protein–protein interaction (PPI) network using the STRING database (https://cn.string-db.org/), prioritizing interactions with a confidence score of at least 0.400. We focused on genes that directly interacted with key hub genes within the network for subsequent analysis. For the visualization and modelling of the PPI network, we utilized Cytoscape software, version 3.9.1. Additionally, we leveraged the GeneMANIA platform (https://genemania.org/) to predict functional associations for our genes of interest and to identify additional hub genes within the interaction network.

## Prediction of RNA–miRNA and mRNA-TF interactions

To identify miRNAs targeting hub genes, we queried the ENCORI database and applied a stringent filter, retaining only those miRNAs corroborated by a minimum of three databases. We then visualized the mRNA–miRNA interaction networks using Cytoscape. For transcription factor (TF) prediction, we consulted the CHIPBase database, prioritizing TFs with substantial support from at least five studies. The mRNA-TF interaction network was subsequently mapped in Cytoscape, revealing a detailed regulatory landscape.

## CIBERSORTx immune infiltration assay

Gene expression matrix data from the UC dataset were uploaded to CIBERSORTx and combined with the LM22 characteristic gene matrix. The data were filtered to exclude any immune cell-enriched fractions below zero, resulting in the acquisition and presentation of specific outcomes in the immune cell infiltration abundance matrix. The stacking histogram displays the distribution of immune cell infiltration proportions among various groups of UC dataset samples. Next, we removed immune cells with an infiltration abundance of 75% or more from the total number of samples. We then presented the variation in immune cell infiltration abundance between the UC and control groups using a chart for group comparisons. Finally, we merged the gene expression matrix of the dataset to construct a correlation heatmap for both the immune cell infiltration abundance and the expression of the hub genes in the UC dataset.

## Construction of diagnostic LASSO and logistic models

To construct a diagnostic model for UC hub genes, we utilized LASSO regression with 10-fold cross-validation on the UC dataset. A seed number of 123 was used, and 1000 cycles were performed to avoid overfitting. Next, we conducted uni-variate analysis using the logistic regression model for the central genes identified by the diagnostic LASSO model. Hub genes were selected at a significance level of 0.10, and those that met the criteria wereused to construct a multivariate logistic model. A nomogram graphically represents the relationship between several independent variables in a rectangular coordinate system using a group of separate line segments. The nomogram illustrated the correlation between the hub genes in the diagnostic model and was created using the results of the multifactor logistic model obtained from the R package rms. It functions as a visualization tool for multivariate logistic regression models and an individualized risk prediction tool. Finally, we generated a calibration curve through calibration analysis to assess the precision and resolution of the logistic model utilizing the hub genes.

## Single-cell data processing

The GSE231993 dataset, comprising single-cell RNA sequencing data, was processed using count data from unique molecular identifiers (UMIs). We pooled all individual samples for comprehensive analysis, we selected 15,000 highly variable genes for principal component analysis (PCA) and subsequent clustering. Data preprocessing, quality control, normalization, and dimensionality reduction clustering were conducted using the Seurat R package. The quality control criteria included the following: expression of each gene in at least 10 cells, a minimum of 200 genes expressed per cell, and maintenance of the proportion of mitochondrial genes below 5%. Default parameters and standard procedures from the Seurat package were used for data normalization, identification of hypervariable genes, and clustering with dimensionality reduction. Data integration across different samples was facilitated using the Harmony R package. A cohesive group of cells was identified using the Seurat package, along with the integration of well-established gene markers and manual annotations.

## UC induction in mice

Female C57BL/6 mice (7−8 weeks old, 18−22 g) were obtained from Spaford (Beijing) Biotechnology Co., Ltd. (licence SCXK [Jing] 2024–0001). The mouse were housed in a specific pathogen-free (SPF) facility under controlled conditions (temperature $22 \pm 1°C$, humidity $55\% \pm 5\%$, and a 12-hour light–dark cycle) and given free access to sterilized chow and water. Mice were acclimated for one week prior to experimentation. All procedures were approved by the Medical Ethics Committee of the First Affiliated Hospital of Yunnan University of Traditional Chinese Medicine (Application No: DW-2024037) and adhered to the ARRIVE guidelines. Mice were randomly assigned to either the normal control (NC) or ulcerative colitis (UC) groups (n = 6 per group). The UC group received 3% dextran sulfate sodium (DSS) in the drinking water for 7 days, and the DSS solution was replaced every 48 hours. On the 8th day, all mice were anesthetized with 1% sodium pentobarbital (3 ml/kg) and euthanized by cervical dislocation. and the entire colon in each mouse was removed from caecum to anus. Colon lengths were measured using a ruler. The section of colon tissues used for Hematoxylin & Eosin was fixed in 4% neutral formalin. and colonic tissues were harvested for subsequent RT-qPCR and western blot analysis.

## Hematoxylin & Eosin (HE)

The colon tissue samples were fixed in 4% neutral formalin then processed for paraffin embedding and sectioning for Hematoxylin & Eosin (HE) staining. The histopathologic change in mucosal structure was observed by hematoxylin and eosin staining under a light microscope and performed Histopathological scores.

 

## Quantitative RT–qPCR

Colonic RNA was extracted using TRIzol (Invitrogen) and reverse transcribed using PrimeScript™ RT Master Mix (TAKARA). RT-qPCR was conducted on an Agilent MX3000P system with SYBR Premix Ex Taq II (TAKARA). The 2-ΔΔCt method was used to calculate. Gene expression relative to that of glyceraldehyde-3-phosphate dehydrogenase (GAPDH) was quantified. The sequences of primers used are listed in Supplementary S3 Table.

## Western blot analysis

Colon tissue was taken out and whose protein content was detected by BCA kit (Beyotime, P0010). Total protein was loaded into SDS-PAGE gel and transferred to the 0.45 µm PVDF membrane. Then the PVDF membrane was washed and blocked in a 5% solution for 1 h at room temperature. After that, the membrane was incubated with the primary antibody of MIF (1:1,000), NAMPT (1:5,000), HIF1A (1:5,000), ACAA2 (1:4,000), PRDX6 (1:4,000), HSPB1 (1:10,000), BNIP3 (1:1,000), PCK2 (1:1,000), All of the above antibodies were purchased from proteintech, and β-actin (Sangon Biotech, JC19KA2355) at 4 °C 1 h, and then incubated with secondary antibody at room temperature for 1 h. The protein bands were imaged and developed with a CLX infrared fluorescence scanning imaging system (Odyssey) and quantified with Image J software.

## Statistical analysis

The statistical analyses were performed using GraphPad Prism (Version 9.0.1) and SPSS (Version 26.0.0.0), and $P < 0.05$ was considered statistically significant. Categorical variables were analysed using the chi-square test or Fisher's exact test. LASSO regression was performed using the glmnet package, and logistic regression was conducted with the rms package. Spearman's correlation was utilized for unspecified analyses to assess correlations between molecules. A P value $\leq 0.05$ indicated statistical significance. For functional enrichment analyses such as GSEA, the Benjamini-Hochberg method is employed for multiple testing corrections, with significance determined by the corrected q-values (FDR).

## Ethics approval and consent to participate

The study was approved by the Medical Ethics Committee of the First Affiliated Hospital of Yunnan University of Traditional Chinese Medicine (Application No: DW2024037) and all animal experiments in this study were conducted in accordance with ARRIVE guidelines for animal reporting. All methods are carried out in accordance with relevant guidelines and regulations.

# Results

## Differential expression of MRGs in UC datasets

The flowchart of this study is presented in Supplementary S1 Fig. To address batch effects, we preprocessed the GSE36807, GSE38713, and GSE47908 datasets by removing batch-related variance, culminating in a consolidated UC dataset. We employed distribution box plots and PCA plots (Fig 1A-1D) to assess the homogeneity of the datasets before and after batch effect mitigation. Postcorrection, the plots confirmed a significant reduction in batch effects among the UC dataset samples.

Within the UC dataset, we identified 21,665 DEGs, among which the UC group exhibited greater expression of a subset of genes than did the control group. Specifically, 1,802 genes met the criteria of |logFC| > 0.5 and $P < 0.05$. In this cohort, 998 genes were upregulated in the UC group (characterized by positive logFC and reduced expression in the control group), while 804 genes were downregulated (marked by negative logFC and increased expression in the control group), we present the difference analysis results as a volcano plot (Fig 2A). Next, we illustrated the DEGs and compared them with the 772 MRGs. This comparative analysis identified 50 significant MRDEGs, with their names listed in S4 Table and visualized through a Venn diagram (Fig 2B).

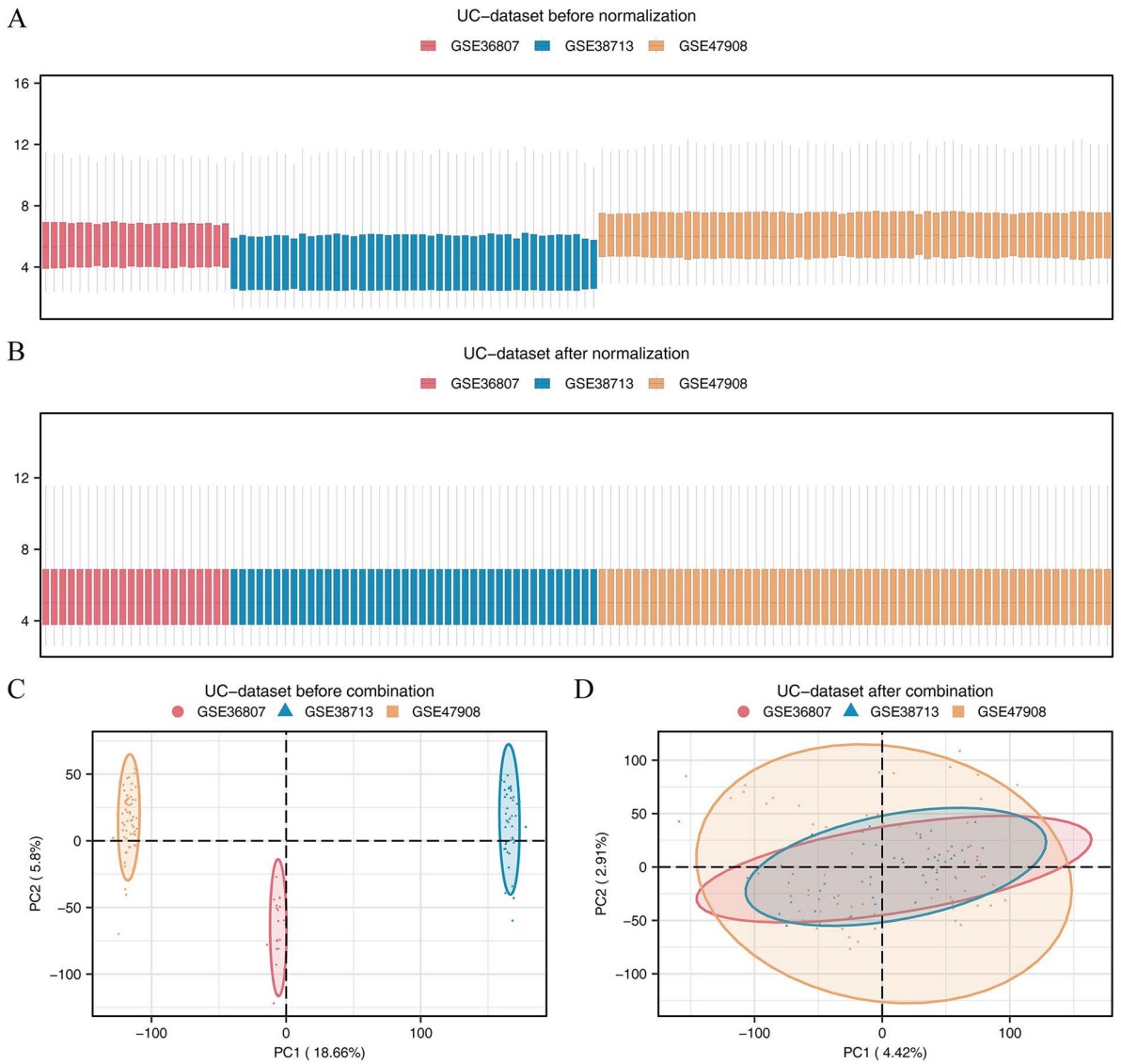

**Fig 1. Differential expression of MRGs in UC datasets.** (A) and after (B) merging. **(C-D)** PCA plots of UC dataset before (C) and after (D) merging.

We subsequently employed a group comparison chart (Fig 2C) to statistically validate the expression variability of these 50 MRDEGs ($P<0.05$) within the UC dataset. The chart revealed a statistically significant disparity in the expression levels of these MRDEGs between the UC and control groups. These genes were then integrated into further analyses. To elucidate the expression patterns, a heatmap was constructed, depicting the differential expression of the 50 MRDEGs in the UC and control groups (Fig 2D).

## Construction of ssGSEA Scores and WGCNA

Using the expression data of the 50 MRDEGs from Supplementary S4 Table, we calculated the MRG scores for each sample in the UC dataset employing the ssGSEA algorithm, which quantifies the mitochondrial response gene (MRG) level. We then applied WGCNA to the samples from the disease group within the UC dataset to identify coexpressed gene

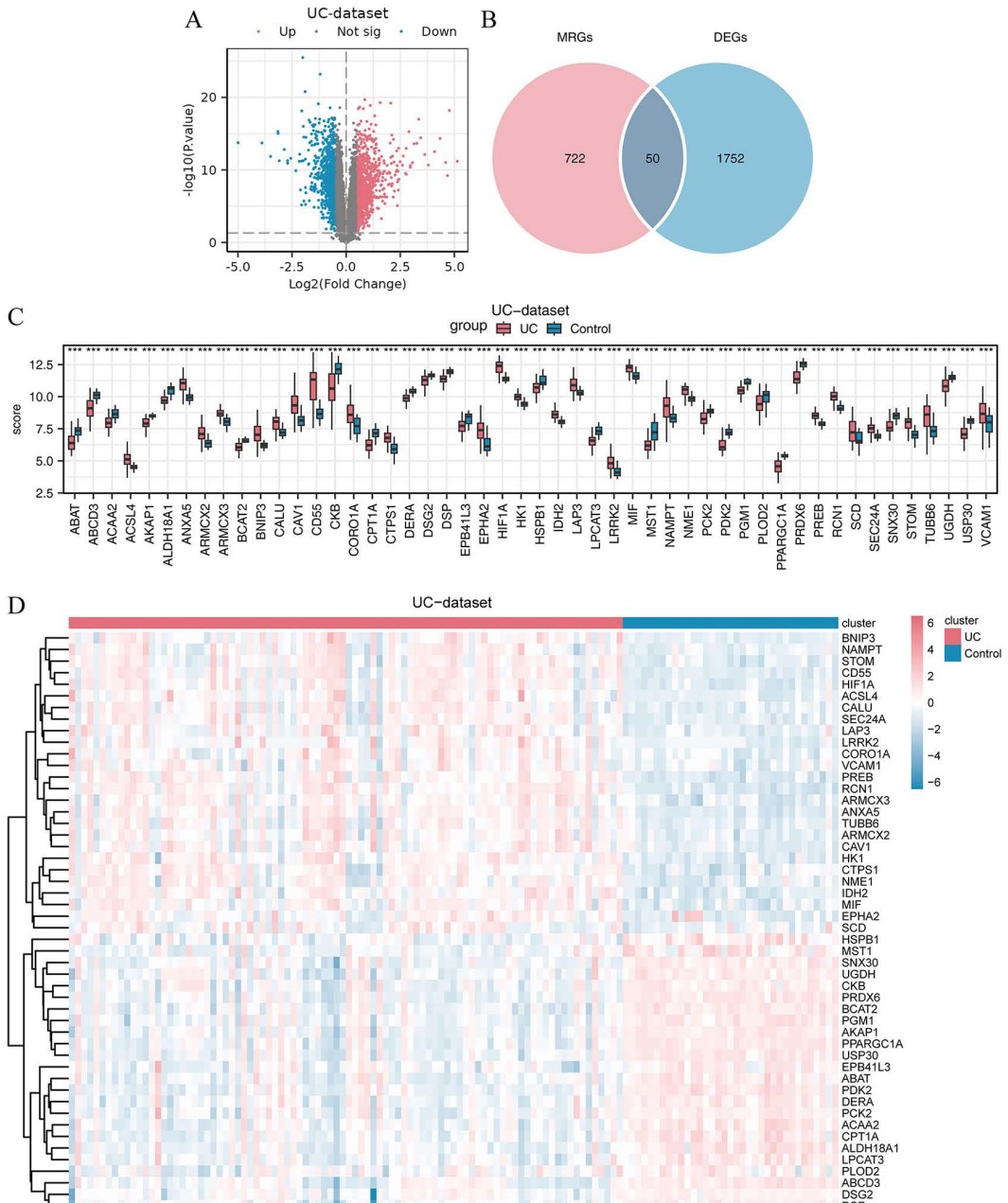

**Fig 2. Differential expression of MRGs in the UC dataset. (A)** Volcano plot of all differential expression analysis between the UC and control groups. **(B)** Venn diagram representation of the intersection of DEGs and MRGs from the UC dataset. **(C)** Comparison of MRDEG expression between the UC and control groups. "score": This score represents the standardized gene expression levels of each mitochondrial autophagy-related differentially expressed gene (MRDEGs) across various samples. **(D)** Heatmap of MRDEGs between the UC and control groups, red scale indicates high expression, white scale indicates moderate expression, and blue scale indicates low expression.

modules. In this analysis, we focused on the top 25% most variable genes for module identification. A clustering tree was constructed to group UC patient samples from the dataset. An optimal power threshold of 0.85 was determined (Fig 3A), with the graph indicating a power of 17 as ideal for WGCNA. Eleven distinct modules emerged from the clustering of the top 25% of variable genes (Fig 3B).

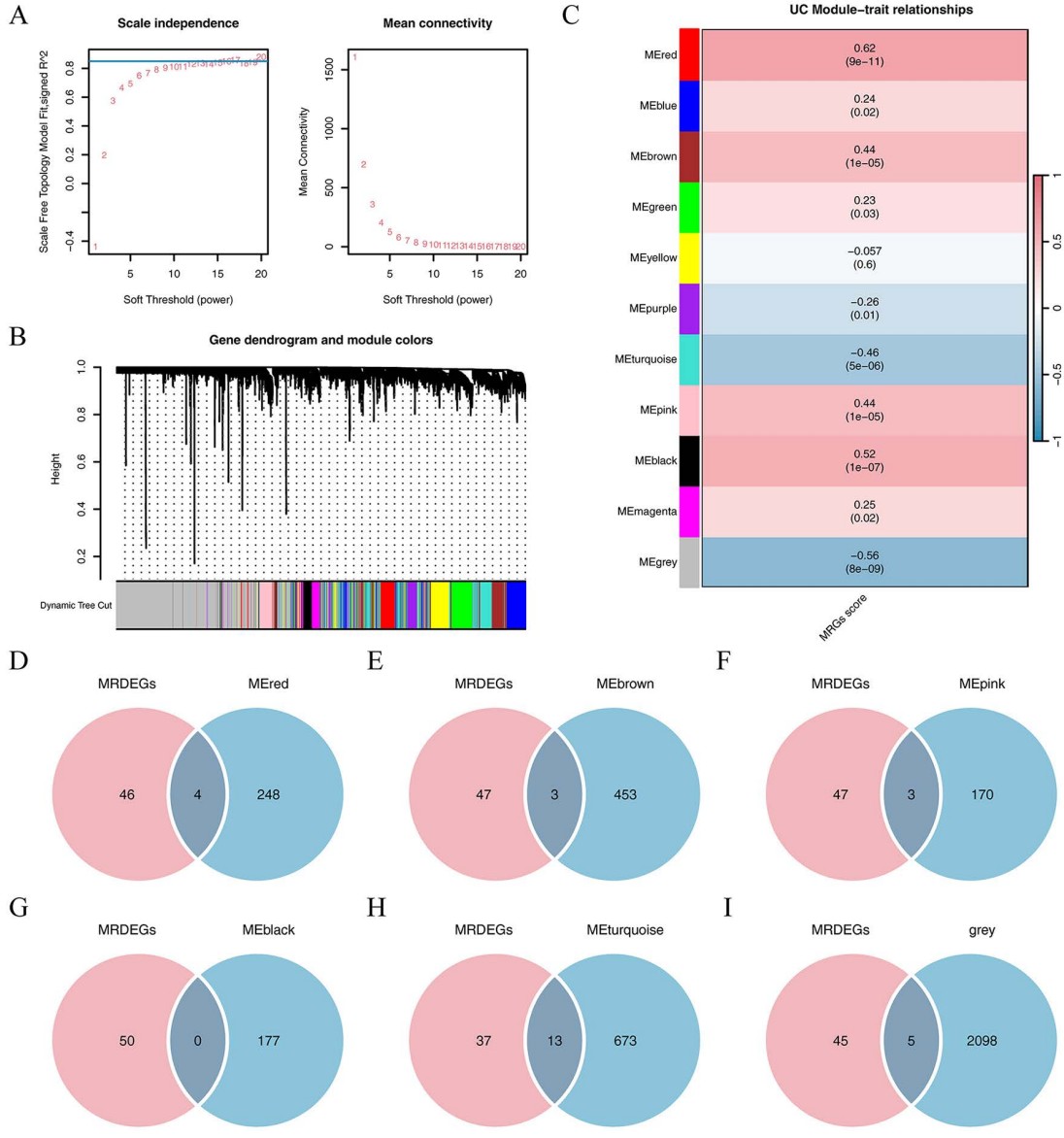

**Fig 3. WGCNA. (A)** Sample module screening threshold scale-free network representation in the UC dataset. **(B)** Correspondence between genes and modules. **(C)** Results of correlation analysis between the gene clustering module and MRG scores are shown. (D-I) Intersection of MRDEGs with the UC dataset of the MEred **(D)**, MEbrown **(E)**, MEturquoise **(F)**, MEpink **(G)**, MEblack **(H)**, and MEblue (I) modules containing module genes is displayed in the Venn diagram.

Subsequently, we visualized the correlation between the top 25% of genes and their respective modules within the UC dataset. For further analysis, we selected genes from the MEred (r = 0.62, P = 9e − 11), MEbrown (r = 0.44, P = 1e − 5), MEturquoise (r = −0.46, P = 5e − 6), MEpink (r = 0.44, P = 1e − 5), MEblack (r = 0.52, P = 1e − 7), and MEblue (r = −0.56, P = 8e − 9) modules that showed a strong correlation (P < 0.05, |r| ≥ 0.30) with the MRG scores of UC patient samples (Fig 3C). We then intersected the 50 MRDEGs with these modules (Fig 3D-3I). Ultimately, 28 hub genes were identified through Venn diagram analysis, focusing on the mitochondrial-autophagy module (ACAA2, ACSL4, AKAP1, ANXA5, BNIP3, CALU, CD55, CKB, CTPS1, EPB41L3, EPHA2, HIF1A, HK1, HSPB1, LAP3, MIF, MST1, NAMPT, NME1, PCK2, PGM1, PLOD2, PPARGC1A, PRDX6, SCD, SEC24A, SNX30, and STOM).

## PPI interaction network and mRNA–miRNA and mRNA-TF prediction networks

The STRING database was used to identify 28 central genes, and an interaction network comprising 20 central genes (ACAA2, ACSL4, ANXA5, BNIP3, CKB, CTPS1, EPHA2, HIF1A, HK1, HSPB1, MIF, NAMPT, NME1, PCK2, PGM1, PPARGC1A, SCD, PLOD2, LAP3, and PRDX6) was constructed (Supplementary S2A Fig). In addition, we used genomic and proteomic data to identify functionally analogous genes via GeneMANIA. Subsequently, we constructed an interaction network involving the 20 hub genes and their functionally similar counterparts (Supplementary S2B Fig).

Using mRNA-miRNA data from the ENCORI database, we predicted miRNAs interacting with hub genes. Subsequently, we screened the candidate miRNAs that supported more than two database entries, and visualized the mRNA-miRNA interaction network using Cytoscape software (S3A Fig). The network comprises 6 mRNA genes (ACADSB, ACSL1, ALDH1A1, ALDH2, ALDH6A1, PC) and 62 miRNAs (specific interactions are detailed in S5 Table).

Using mRNA-TF data from the ChIPBase3.0 database, we predicted transcription factors (TFs) interacting with hub genes. Subsequently, we screened the TFs with more than 10 supporting samples and visualized the mRNA-TF interaction network using Cytoscape software (S3B Fig). The network comprises 16 mRNA genes (ACADSB, ACOX2, ACSL1, ADH1B, ADH1C, ADH4, ADH6, ALDH1A1, ALDH1L1, ALDH2, ALDH6A1, ALDH8A1, CYP4A11, EHHADH, PC, PFKFB1) and 38 TFs (specific mRNA-TF interactions are detailed in S6 Table).

## Correlation analysis of the hub genes

We examined the correlations among the expression of 20 pivotal genes (ACAA2, ACSL4, ANXA5, BNIP3, CKB, CTPS1, EPHA2, HIF1A, HK1, HSPB1, MIF, NAMPT, NME1, PCK2, PGM1, PPARGC1A, SCD, PLOD2, LAP3, and PRDX6) across all disease group samples in the UC dataset using the Spearman correlation algorithm. The correlation matrix was visualized through a chord diagram (Fig 4A) and a heatmap (Fig 4B), revealing three distinct clusters of genes based on their expression patterns. PLOD2 displayed minimal significant correlations with the other genes. The second cluster included ACAA2, CKB, PCK2, PGM1, PPARGC1A, and PRDX6, while the third was composed of ACSL4, ANXA5, BNIP3, CTPS1, EPHA2, HIF1A, HK1, HSPB1, MIF, NAMPT, NME1, SCD, and LAP3. A robust positive correlation was identified between the second and third clusters, whereas a notable negative correlation was observed between the first and third clusters.

To evaluate the functional relatedness of the hub genes, we performed a functional similarity analysis, which is graphically represented as a box plot (Fig 4C). NME1 exhibited the highest score for functional similarity. We selected the top four gene pairs with the strongest positive correlation coefficients from the heatmap for further visualization using a scatter plot (Fig 4D-4G). In the UC dataset, significant positive correlations were identified between NAMPT and HIF1A ($r = 0.87$, $P < 2.2e - 16$; Fig 4D), between PRDX6 and CKB ($r = 0.87$, $P < 2.2e - 16$; Fig 4E), between HIF1A and ACSL4 ($r = 0.82$, $P < 2.2e - 16$; Fig 4F), and between HIF1A and ANXA5 ($r = 0.82$, $P < 2.2e - 16$; Fig 4G).

## CIBERSORTx immune infiltration assay

Employing the CIBERSORTx algorithm, we initially utilized stacked histograms to ascertain the proportion of immune cell infiltration across samples (Fig 5A), excluding those immune cells with negligible infiltration. A comparative chart was then generated to illustrate variations in immune cell infiltration abundance as determined by the CIBERSORTx algorithm (Fig 5B), with cells not exceeding 75% of the total sample infiltration abundance removed from further analysis. The analysis revealed significant differences in the infiltration of 12 immune cell types (T cells CD8, T cells CD4 memory resting, T cells CD4 memory activated, T cells regulatory (Tregs), NK cells resting, Macrophages M0, Macrophages M1, Macrophages M2, Dendritic cells resting, Mast cells resting, Mast cells activated, and Neutrophils) between the UC and control groups ($P < 0.05$)

Subsequently, a correlation heatmap was constructed to visualize the immune cell abundance (Fig 5C-5D). A heatmap correlating immune cells with the 20 hub genes (ACAA2, ACSL4, ANXA5, BNIP3, CKB, CTPS1, EPHA2, HIF1A, HK1, HSPB1, MIF, NAMPT, NME1, PCK2, PGM1, PPARGC1A, SCD, PLOD2, LAP3, and PRDX6) (Fig 5D) revealed strong

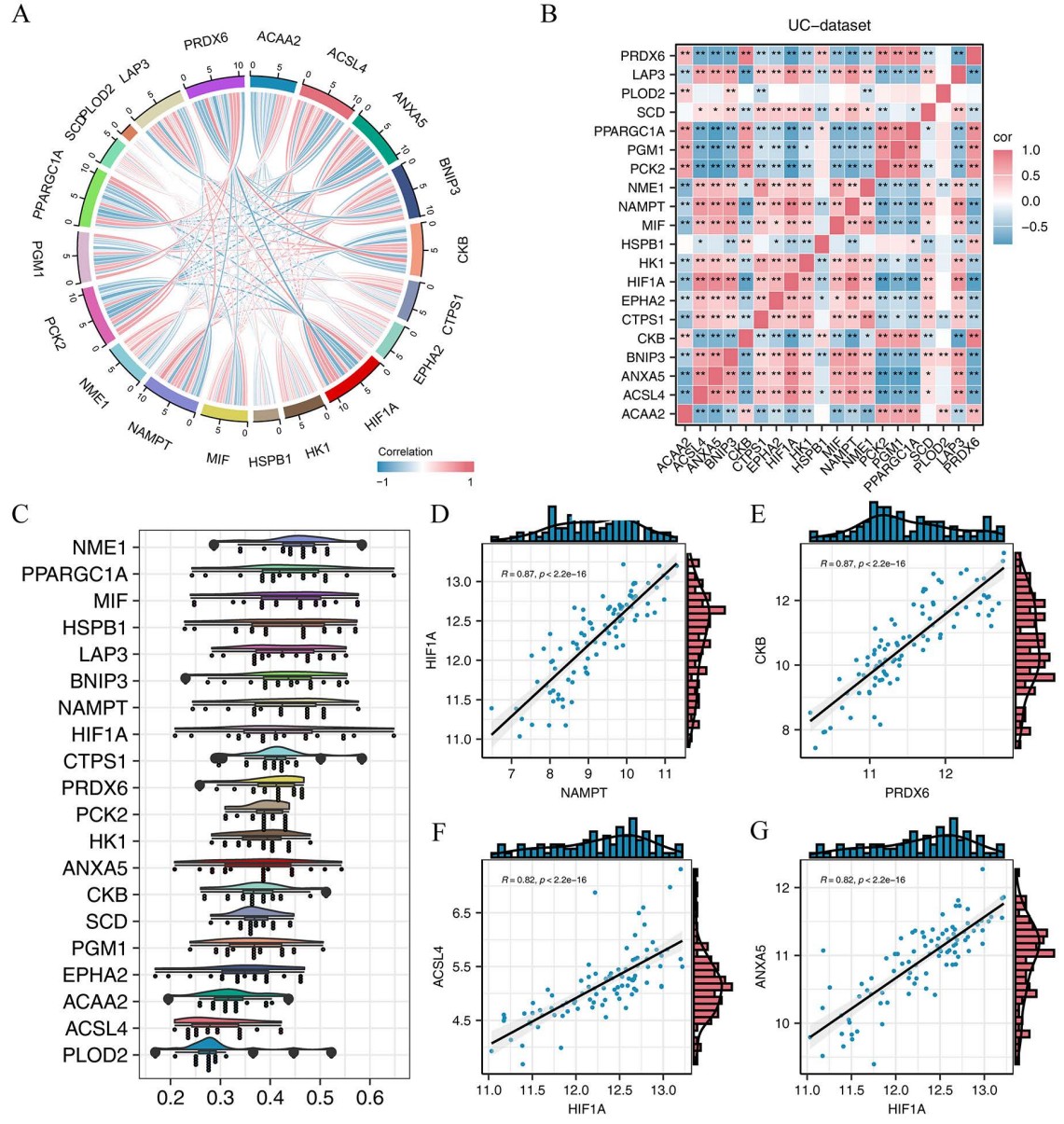

**Fig 4. Correlation analysis of hub genes in UC dataset. (A-B)** Displayed are correlation chord graph (A) and correlation heat map (B) illustrating hub genes in UC dataset. **(C)** Box plot displaying scores of functional similarity for central genes. **(D)** Scatter plot illustrating correlation between NAMPT gene and HIF1A gene. **(E)** Scatter plot illustrates relationship between genes PRDX6 and CKB. **(F)** Scatter plot illustrating relationship between genes HIF1A and ACSL4. **(G)** Scatter plot displays relationship between genes CYP4A22 and ADH6.

positive correlations (r > 0, P < 0.05) between M1 macrophages and activated CD4 + memory T cells, as well as between resting mast cells and a subset of immune cells, including M0 and M2 macrophages and resting DCs. Additionally, a notable positive correlation was observed between neutrophils and activated immune cells (activated CD4 + memory T cells, M0 macrophages, M1 macrophages, and activated mast cells). Most of the hub genes showed a robust positive correlation (r > 0, P < 0.05) with activated immune cells and a significant negative correlation (r < 0, P < 0.05) with the levels of resting immune cells (CD8 + T cells, Tregs, M2 macrophages, resting DCs, and resting mast cells).

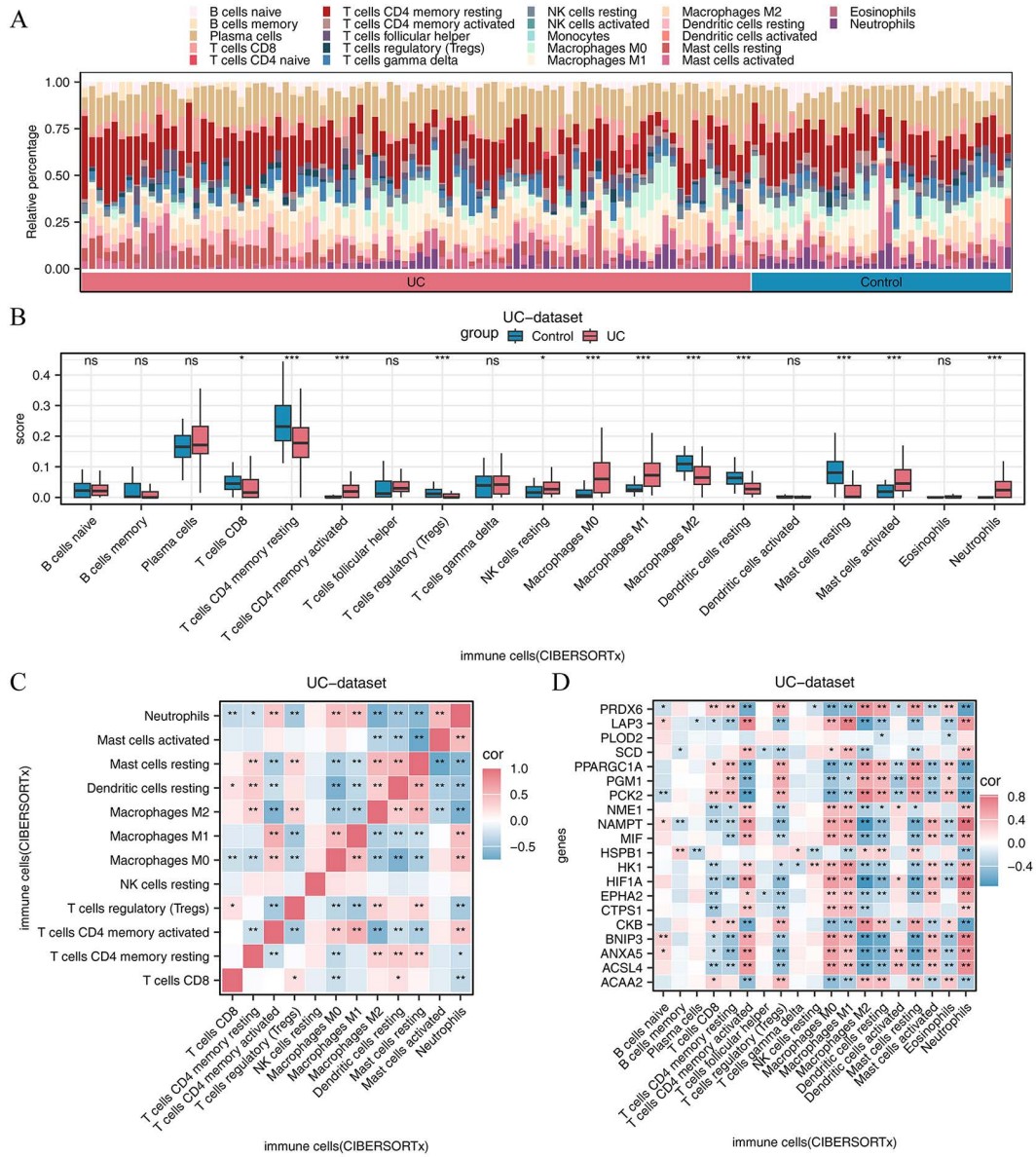

**Fig 5. CIBERSORTx immune infiltration assay. (A)** Stacked histograms of immune cells in different samples of UC dataset under CIBERSORTx algorithm. Bars of different colors represent different immune cells. **(B)** Group comparison of infiltration abundance of immune cells in UC dataset under CIBERSORTx algorithm between UC and control groups. **(C)** Heat map of correlation of infiltration abundance between immune cells in UC dataset under CIBERSORTx algorithm. **(D)** Heat map of correlation between immune cell infiltration abundance and hub gene expression in UC dataset under CIBERSORTx algorithm. Asterisk in correlation heat map.

## Construction of diagnostic LASSO and logistic regression models

To evaluate the diagnostic potential of the 20 central genes (ACAA2, ACSL4, ANXA5, BNIP3, CKB, CTPS1, EPHA2, HIF1A, HK1, HSPB1, MIF, NAMPT, NME1, PCK2, PGM1, PPARGC1A, SCD, PLOD2, LAP3, and PRDX6) within the UC dataset, we constructed a diagnostic model utilizing LASSO regression based on gene expression levels (Fig 6A). The LASSO path plot (Fig 6B) visualized the relationship between the genes and the model's predictive power. This analysis

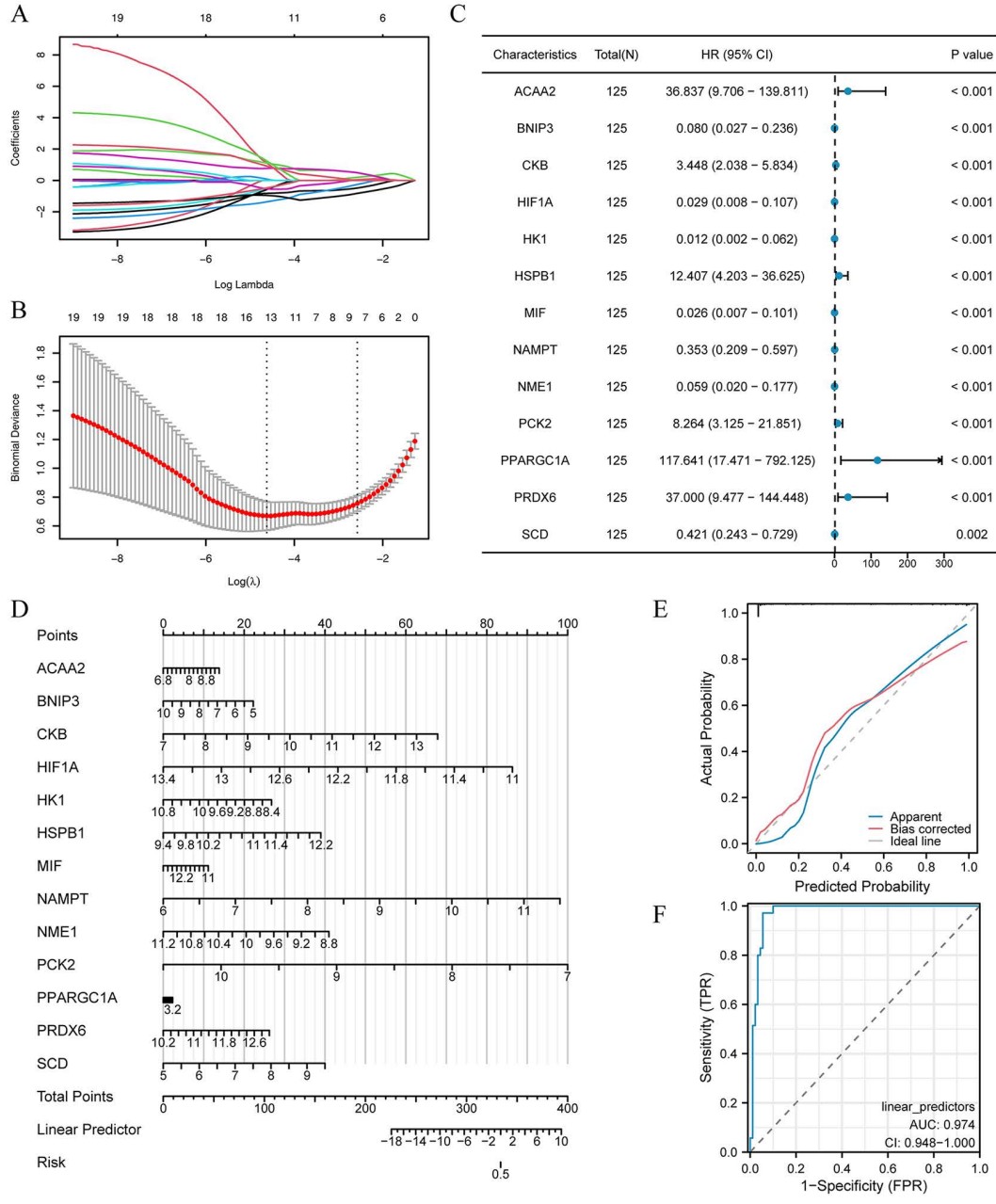

**Fig 6. Construction of univariate and multivariate Cox model. (A)** LASSO regression diagnostic model diagram for hub genes. **(B)** Variable trajectory plot of LASSO regression diagnostic model. **(C)** Forest plot and **(D)** diagnostic nomogram presentation of multivariate logistic model. When the gene expression value exceeds the scale range of the column chart, the maximum or minimum value of the axis should be taken to ensure the continuity and stability of the model application. **(E)** Presentation of multivariate logistic model includes diagnostic calibration curve and **(F)** ROC curve.

culminated in a model that included 13 hub genes: ACAA2, BNIP3, CKB, HIF1A, HK1, HSPB1, MIF, NAMPT, NME1, PCK2, PPARGC1A, PRDX6, and SCD. A univariate logistic diagnostic model, with a threshold of 0.1, successfully identified all 13 central genes (Fig 6C, Supplementary S7 Table).

We then developed a multivariate logistic model incorporating these 13 genes using the GDM dataset. The resulting model is presented with a diagnostic nomogram (Fig 6D), which aids in the interpretation of the model's predictive capabilities. A diagnostic calibration curve (Fig 6E) and ROC curve (Fig 6F) were generated to assess the model's performance. The ROC curve analysis indicated a close alignment of the prediction curve (blue line) with the ideal curve (gray line), signifying an excellent model fit. The model's diagnostic efficacy was further confirmed by an AUC of 0.974, as demonstrated by the ROC curve.

## Quality control of single-cell sequencing data

To analyse the MRGs in patients with UC, we obtained the single-cell sequencing dataset GSE231993 to assess their expression and significance. After filtering, 15478 cells were obtained based on the number of gene features, gene counts, and proportion of mitochondrial genes (Fig 7A). Following data normalization, 15000 highly variable genes were selected to reduce dimensionality using PCA (Fig 7B). Subsequently, we employed the R package Harmony to eliminate batch effects. We observed a strong correlation and effective quality control in the scatter plot displaying the relationship between nCount and nFeatures (Fig 7C). The single-cell dimensionality reduction diagram (Fig 7D-7E) illustrated that each cell population was dispersed among distinct clusters and samples, demonstrating the impact of batch removal.

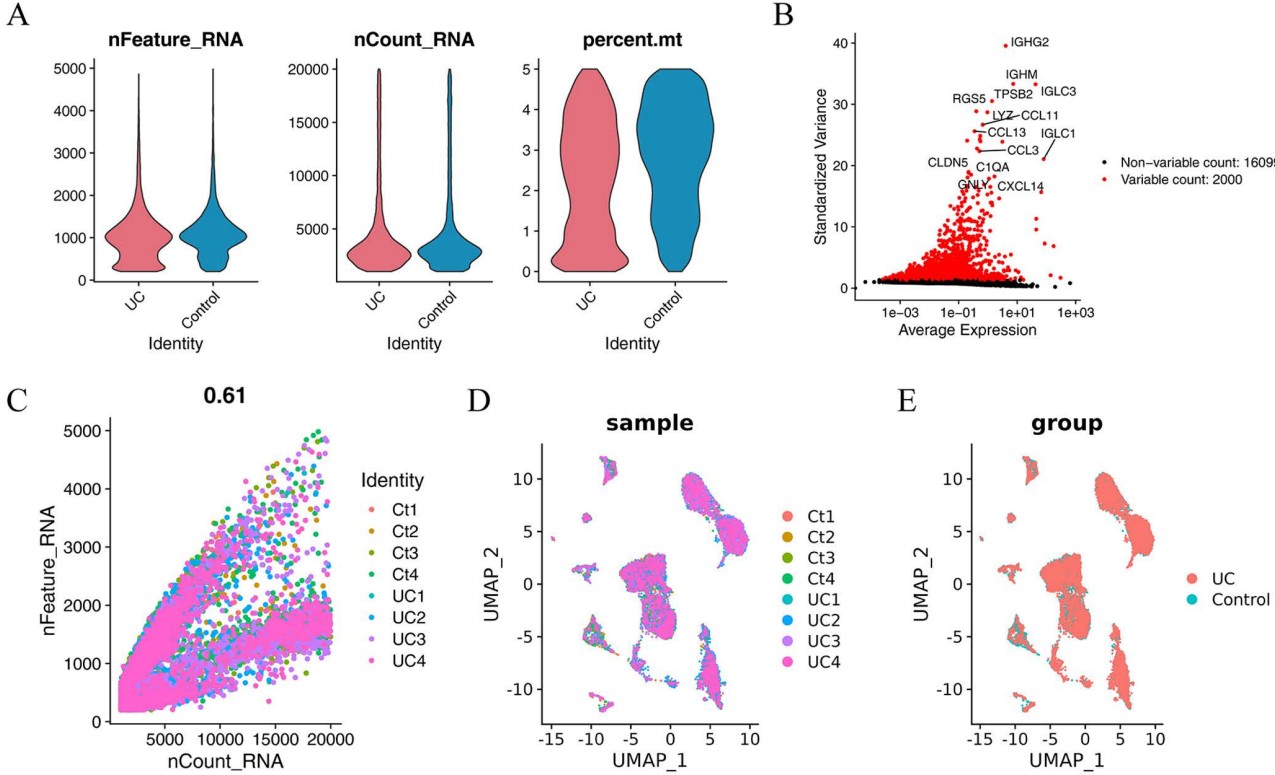

**Fig 7. Single-cell sequencing data quality control. (A)** Violin plot of count of genes, number of sequences, and mitochondrial proportion in sample. **(B)** Scatter plot of hypervariable genes, the visualization only displays the top 2000 highly variable genes, which is the default setting of Seurat package for high variability gene visualization, and is only used for visual display. **(C)** Scatter plot of correlation between number of sequences and number of genes in all cells. **(D)** UMAP plots show single-cell dimensionality reduction effect of different samples from patients with UC, differentiated by individual. **(E)** UMAP plots show effect of single-cell dimensionality reduction on diseased and diseased tissue samples from patients with UC, differentiated by patient grouping.

## Cell type identification from single-cell sequencing data

Using the R package singleR, 20 cell clusters were detected (Fig 8A). This information was derived from genes that indicate the type of cell (Fig 8C) and a combined count of seven different cell types (Fig 8B), including fibroblasts, B cells, monocytes, epithelial cells, CD8+T cells, endothelial cells, and haematopoietic stem cells (HSCs). We confirmed

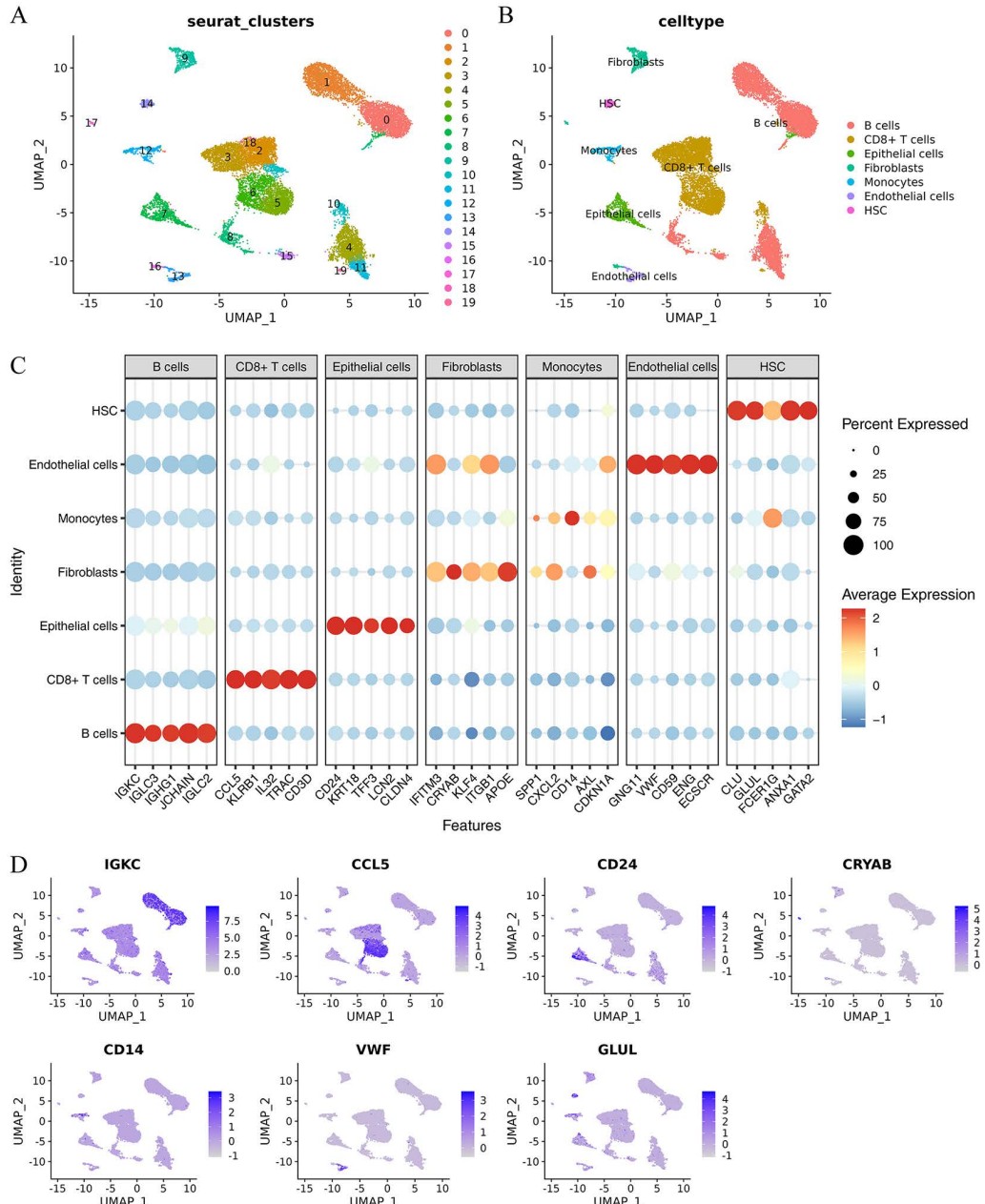

**Fig 8. Process of determining cell type in single-cell sequencing data. (A)** Twenty cell populations were created from GSE231993 dataset. **(B)** Twenty groups of cells were categorized into seven distinct cell types based on markers specific to each cell type. **(C)** Expression of specific markers for different cell types in 20 cell clusters. Color intensity indicates gene expression level, while dot size reflects gene's positive proportion within cell. **(D)** UMAP feature plot displaying expression of marker genes for different cell types.

the expression patterns of seven marker genes in the single-cell UMAP space using UMAP feature plots. Our findings revealed that distinct cell types exhibited specific expression of these markers (Fig 8D).

### Difference analysis of cell type grouping in single-cell sequencing data

In the GSE231993 dataset, we performed differential expression analyses for each cell type to identify unique gene expression patterns. Utilizing a significance threshold of |log2FC| > 1 and adjusted p value (p.adj) < 0.05, we highlighted genes with the most substantial positive and negative log2FC values (Fig 9A). For B cells, the top genes with positive and negative log2FC values included IGLC3, IGKC, IGLC1, JCHAIN, IGHG1, IL32, KLRB1, ANXA1, CCL5, and TPSB2. Among CD8+T cells, CD3D, IL32, CCL5, TRAC, KLRB1, IGLC3, IGKC, IGHG1, IGLC1, and IGLC2 were the most differentially expressed. The top genes related to endothelial cells were PLVAP, SPARCL1, CD320, FABP5, CLDN5, JCHAIN, and IGLC1. The most significantly differentially expressed genes were PHGR1, LGALS4, C15orf48, FABP1, KRT8, CCL5, CD74, CREM, SRGN, and CXCR4 in epithelial cells. Fibroblasts were characterized by APOE, CFD, DCN, C1S, IGFBP7, JCHAIN, IGKC, IGHG1, IGHG3, and IGLC1, while HSCs displayed TPSB2, TPSAB1, CTSG, CPA3, CLU, CXCR4, JCHAIN, IGKC, and IGLC1. Monocytes uniquely expressed LYZ, CST3, AIF1, TYROBP, CXCL8, JCHAIN, IGHG1, IGKC, IGHA1,and IGLC2 and were identified as CD74.

From the differential analysis of GSE231993, we identified and labelled 13 hub genes (ACAA2, BNIP3, CKB, HIF1A, HK1, HSPB1, MIF, NAMPT, NME1, PCK2, PPARGC1A, PRDX6, and SCD) within our logistic model. Analysis (Fig 9B) revealed that HSPB1 was significantly upregulated in endothelial and fibroblasts (log2FC > 0, p < 0.05). CKB showed elevated expression in epithelial cells, PRDX6 in HSCs, and NAMPT in monocytes. Additionally, we quantified the cellular composition in each sample (Fig 9C) and across different sample groups (Fig 9D) and observed no significant differences.

### Confirmation of the Hub Genes in the Mice Used as Model for Colitis

Ultimately, as shown in Fig 10, we validated the gene expression patterns of hub genes in a murine model of DSS induced acute colitis. There are significant weight loss and colon shortening in DSS-induced colitis, indicating a successful model was established (Fig 10A-10C). Furthermore, after hematoxylin-eosin staining, in the UC group, the typical structure of the mucosa is destroyed, the crypt structure is indistinct, goblet cells disappear, a large number of inflammatory cells infiltrate, and bleeding spots appear in the mucosa of the colon (Fig 10D). The histological score of the DSS group exhibited an obviously higher level compared to that of the other group (Fig 10E). The disease activity index (DAI) of the DSS group was notably worse compared to the control group (Fig 10F)

### Validation by qRT-PCR

Quantitative real-time PCR (qRT-PCR) confirmed that the mRNA levels of BNIP3, NAMPT, MIF, and HIF1A were elevated in the UC group compared to the control group, while the levels of PRDX6, ACAA2, and HSPB1 were reduced. These findings align with our bioinformatic analysis. However, the expression levels of HK1, SCD, CKB, NME1, and PPARGC1A did not significantly differ between the two groups (Fig 11). This discrepancy may stem from variations in sample sizes between the dataset and the experimental cohort.

### Western blot analysis

Further, compared to NC, the protein expression of MIF, NAMPT, BNIP3 and were significantly up-regulated in UC (p < 0.05, p < 0.01 or p < 0.001), ACAA2, HSPB1, PRDX6 and PCK2 Protein expression were significantly reduced (p < 0.05, p < 0.01 or p < 0.001) Fig 12, full blot in S4 Fig).

### Discussion

Ulcerative colitis (UC) is a chronic inflammatory bowel disease that significantly impacts patients' quality of life and is often accompanied by extraintestinal manifestations [7–9]. The exact pathogenesis of UC remains incompletely understood,

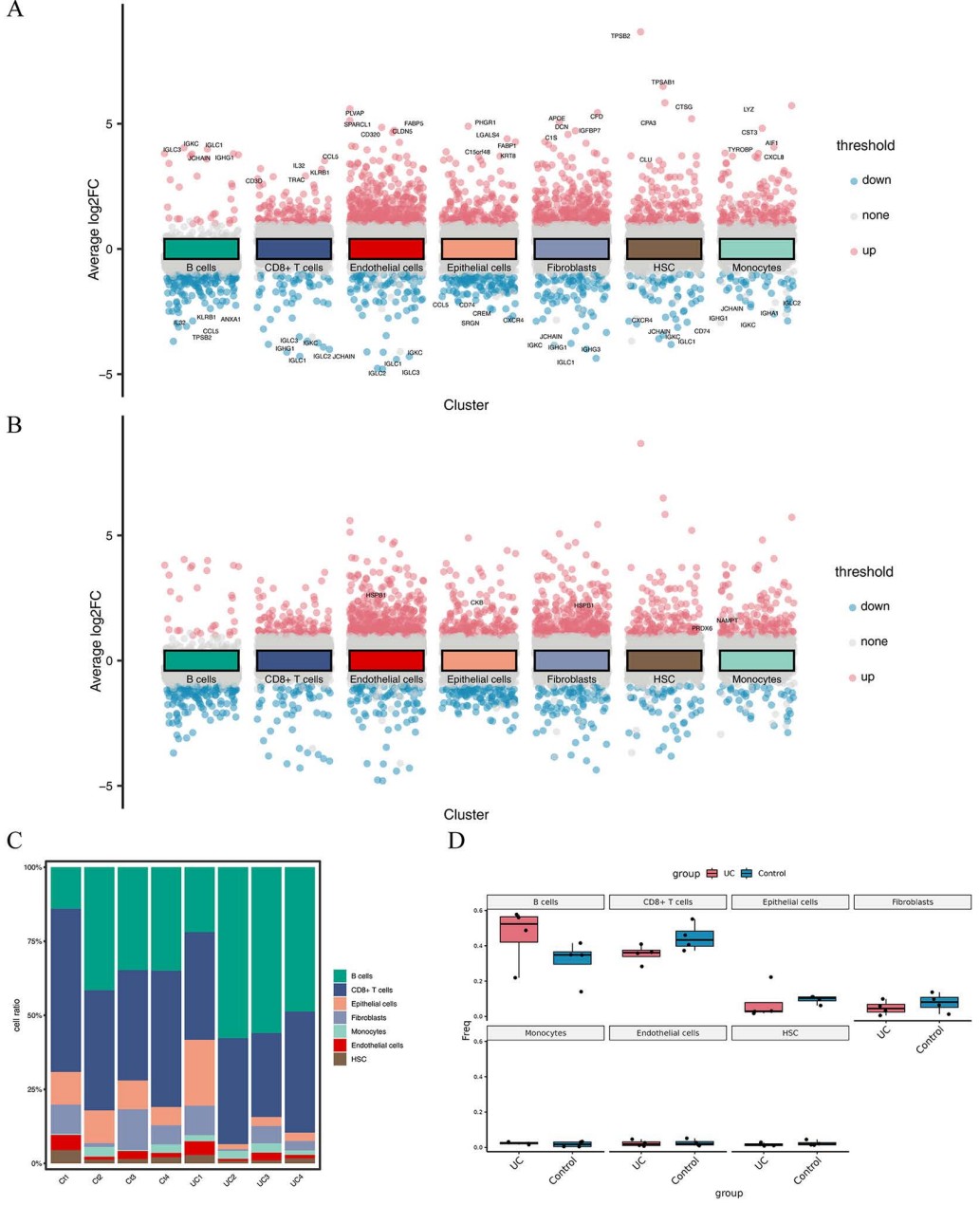

**Fig 9. Analysis of variation in cell type grouping in single-cell sequencing data. (A)** Differential analysis of celltype groupings in GSE231993 dataset displayed positive and negative top5 genes for marker log2FC. **(B)** Differential analysis of cell type groupings in GSE231993 dataset displayed positive and negative top 5 genes for marker log2FC. **(C)** Bar plot of ratio of cell types in every sample in GSE231993 dataset. **(D)** Box plot of ratio of cell types between UC and control groups in GSE231993 dataset.

but immune dysregulation, genetic susceptibility, and mitochondrial dysfunction are considered closely associated with its onset and progression [7,10]. Mitochondrial autophagy, as a critical process for maintaining mitochondrial quality control and cellular homeostasis, plays a pivotal role in inflammation regulation, oxidative stress, and mucosal barrier homeostasis [11–13]. Therefore, systematic identification of key molecules related to mitochondrial autophagy in UC is of great significance for deepening the understanding of disease mechanisms and screening potential biomarkers.

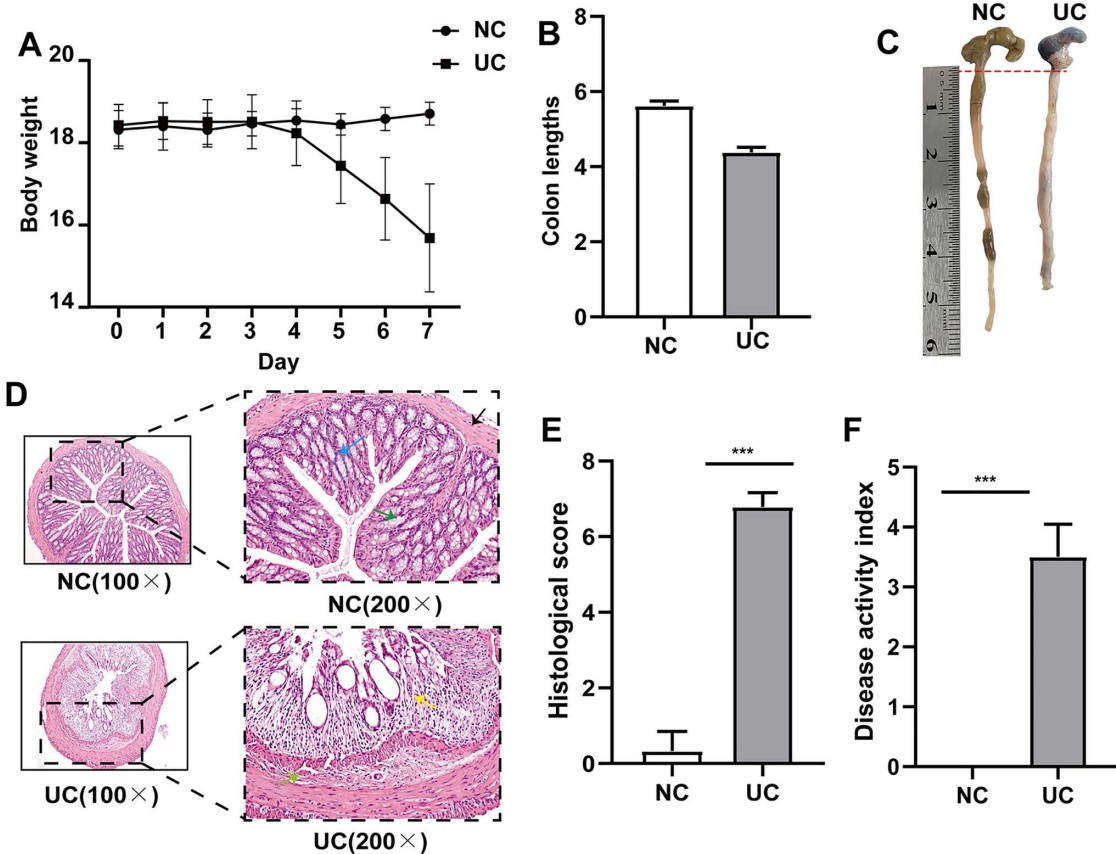

**Fig 10. DSS-induced colitis mice model was successfully established. (A)** Body weight, n = 6 in each group. **(B)** Length of colons. **(C)** Colon appearance. **(D)** H&E-stained colon tissues. **(E)** Histological score. **(F)** Disease activity index. The results are presented as the means ± SD for six samples/group. Blue arrow: crypt structure; green arrow: goblet cells; black arrow: mucosa of the colon; red arrow: mucosal bleeding spots; yellow arrow: inflammatory cells;*$p < 0.05$, **$p < 0.01$ and ***$p < 0.001$ compared with the control group (NC).

This study systematically screened key mitochondrial autophagy-related genes associated with ulcerative colitis (UC) based on integrated transcriptomic and single-cell transcriptomic analyses, and further constructed a diagnostic model. The results demonstrated that multiple mitochondrial autophagy-related genes exhibit abnormal expression in UC, with some key genes showing significant correlation with immune cell infiltration and demonstrating distinct cell-type-specific expression patterns. These findings suggest that mitochondrial autophagy dysfunction may not only contribute to metabolic imbalance in UC but also be closely linked to the remodeling of the intestinal inflammatory microenvironment.

Among the identified hub genes, ACAA2, BNIP3, and NAMPT may be particularly associated with ulcerative colitis (UC). ACAA2 is involved in fatty acid oxidation and may be implicated in metabolic abnormalities and immune alterations during intestinal inflammation [14]. BNIP3, a classical mitochondrial autophagy regulator, may influence UC progression through mitochondrial quality control and inflammatory signaling [15–17]. NAMPT participates in energy metabolism and immune regulation and has been implicated in intestinal inflammation [18–21]. These findings suggest that metabolic dysregulation related to mitochondrial autophagy may contribute to the pathogenesis and progression of UC, although the specific mechanisms require further validation.

The regulatory network analysis we constructed further suggests that miRNAs and transcription factors may be involved in the regulation of mitochondrial autophagy-related genes in UC. Among them, miR-224-5p and miR-20b-5p

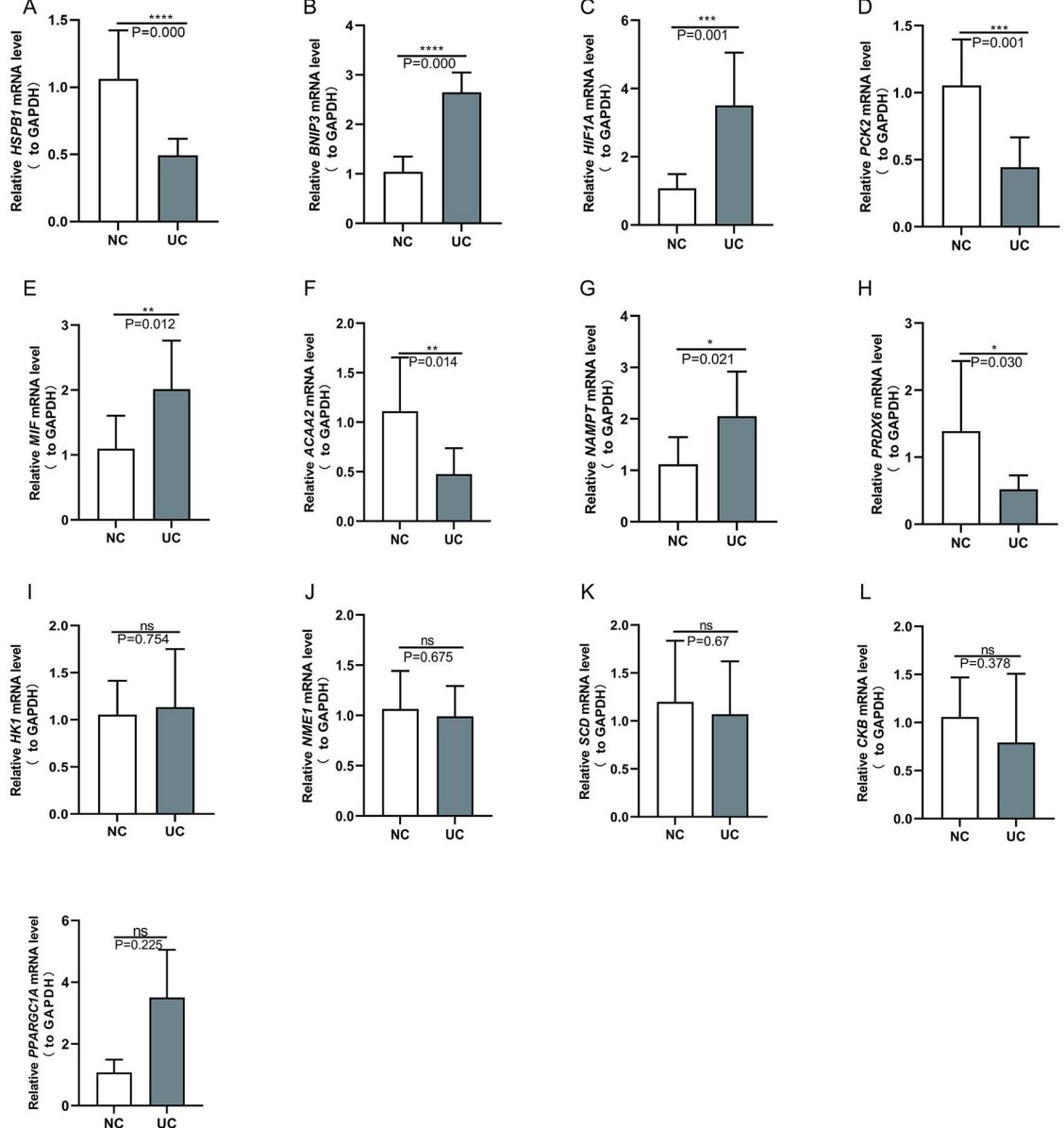

**Fig 11. Verification of the 13 diagnostic hubgenes using real-time quantitative PCR (UC samples = 6, control samples = 6).** *$p < 0.05$, **$p < 0.01$ and ***$p < 0.001$.

have been reported to be associated with inflammatory responses, apoptosis, and macrophage polarization [22–25]; CREB1 and GRHL2 are implicated in intestinal inflammation and epithelial barrier function regulation [26–29]. These findings indicate that upstream post-transcriptional and transcriptional level regulation may contribute to abnormalities in mitochondrial autophagy-related pathways. However, since the correlation analysis in this study was primarily based on database predictions, their biological significance still requires further confirmation through subsequent functional experiments.

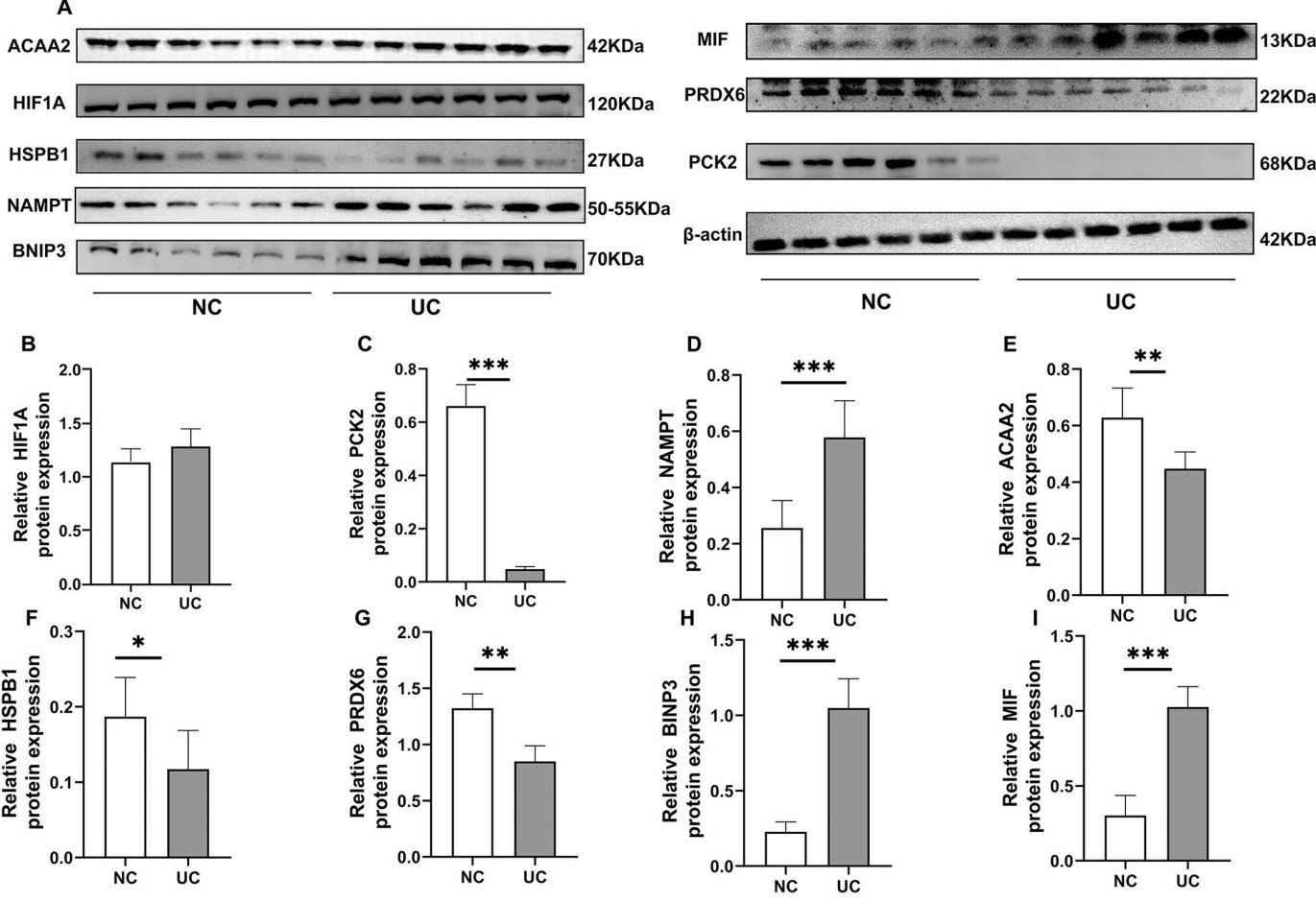

**Fig 12. Using Western blotting to validate eight significantly changed genes.** Original blots are presented in Supplementary S4 Fig. The quantitative analysis of MIF protein expression was performed using the monomer bands shown below. (UC samples = 6, control samples = 6). *$p < 0.05$, **$p < 0.01$ and ***$p < 0.001$.

Immunohistochemical infiltration analysis revealed significant alterations in the proportions of multiple immune cell types in UC, particularly involving macrophages, T cells, NK cells, and dendritic cells. Abnormal activation of M1 and M2 macrophages may exacerbate inflammatory responses by producing pro-inflammatory cytokines [30,31]. Additionally, the role of dendritic cells (DCs) and NK cells in UC cannot be overlooked, as they play critical roles in shaping adaptive immune responses and maintaining immune tolerance [32–35]. Furthermore, most hub genes exhibited significant correlations with distinct immune cell subpopulations, suggesting that mitochondrial autophagy-related molecular changes may be closely associated with immune activation states and disruption of immune homeostasis. Single-cell analysis results further demonstrate that these molecular changes are not uniformly distributed across all cells but exhibit specific cell population biases, indicating pronounced spatial and cellular heterogeneity in the inflammatory response of UC.

In terms of diagnostic value, the mitochondrial autophagy-associated polygenic model constructed in this study demonstrated good discriminative ability within the internal dataset, achieving an ROC-AUC of 0.974, suggesting its potential for distinguishing ulcerative colitis (UC) from control tissues. However, these results should be interpreted with caution. Firstly, the public dataset used lacked detailed clinical information such as disease activity levels, endoscopic scores, medication history, and long-term follow-up data, making it impossible to further evaluate the model's value in disease stratification, prognosis assessment, or

treatment response prediction. Secondly, this study only validated the gene set's discriminative capacity between UC and relatively normal colonic tissues, without systematic evaluation in more clinically challenging differential diagnostic scenarios, such as comparisons with Crohn's disease or other inflammatory bowel diseases. Therefore, at this stage, the gene panel should be regarded as an exploratory molecular classification feature rather than a direct mature clinical diagnostic tool.

Single-cell transcriptome analysis further elucidated the cellular origins of certain key genes. The results demonstrated that HSPB1 was predominantly enriched in endothelial cells and fibroblasts, PRDX6 exhibited higher expression levels in HSCs [36,37], CKB was primarily localized in epithelial cells [38,39], while NAMPT was mainly distributed in monocytes [40]. These findings suggest that distinct mitochondrial autophagy-related genes may perform diverse functions across different cell populations, involving processes such as epithelial metabolic regulation, stress response, tissue repair, and immune modulation. Therefore, single-cell-level localization of key genes facilitates a more accurate understanding of their potential roles in the UC microenvironment.

In the animal experiment section, multiple findings in the DSS-induced colitis model were largely consistent with trends observed in human transcriptome analysis. qRT-PCR results demonstrated upregulation of BNIP3, NAMPT, MIF, and HIF1A in the model group, while downregulation of ACAA2, PRDX6, and HSPB1 was noted. Western blot results similarly revealed consistent changes at the protein level for several key molecules. These findings provide preliminary in vivo evidence for the involvement of these genes in intestinal inflammation and mitochondrial stress processes. However, it should be emphasized that the DSS model primarily reflects biological changes in an inflammatory environment and cannot serve as an external validation alternative for independent clinical populations. Therefore, animal experiment results are more suitable for supporting the biological rationale of candidate molecules rather than directly demonstrating the generalizability of model clinical applications.

The integration of bulk RNA-seq and scRNA-seq technologies can provide complementary insights into disease mechanisms, yet there remains no universally accepted analytical workflow [41–44]. This study adopted a holistic-to-localized analytical strategy: first screening robust candidate genes from bulk transcriptomic data, then utilizing single-cell data to map their expression across distinct cell populations. This design effectively balances the relative stability of bulk data with the cellular resolution of scRNA-seq, while minimizing biases introduced by platform variations.

Therefore, in this study, we adopted a strategy of "first conducting robust candidate gene screening at the tissue level using bulk data, followed by cell type localization and refined interpretation of expression patterns with single-cell data" to maximize the advantages of sample size and cellular resolution. Specifically, at the bulk level, we combined multiple independent cohorts, performed rigorous batch effect correction (sva/Combat), differential analysis, and WGCNA joint screening, and cross-validated with an independent mitochondrial autophagy gene set to identify candidate genes. This approach enhances signal stability across larger sample sizes and reduces the impact of single-cohort or platform biases. At the scRNA-seq level, we utilized Harmony for cross-sample integration, performed cell annotation based on SingleR combined with established literature markers, and primarily focused on the relative enrichment and localization of the aforementioned candidate genes within major cell populations, rather than "discovering new genes" from single-cell data itself, thereby minimizing bias caused by platform differences and annotation uncertainties.

This study still has several limitations. First, all analyses were based on colon tissue data and did not include non-invasive samples such as blood or feces, so its generalization to real-world clinical testing scenarios requires caution. Second, the lack of comprehensive clinical information in public databases prevented further evaluation of the clinical significance of these molecules from perspectives such as disease activity, mucosal healing, pharmacotherapy, or long-term prognosis. Third, the gene panel in this study was validated only in the merged internal cohort and the DSS model, lacking independent external cohort and multicenter population data support. Finally, the study results primarily reflect expression correlations and overall trends, which are insufficient to fully elucidate the causal relationships between mitochondrial autophagy abnormalities, immune dysregulation, and mucosal injury. Further in-depth research combining prospective clinical cohorts and mechanistic experiments is warranted.

Overall, this study systematically analyzed the abnormal expression characteristics of mitochondrial autophagy-related genes in ulcerative colitis (UC) at both tissue level and single-cell level, identified multiple key molecules with potential significance, and constructed a polygenic model with certain discriminative capacity. The results suggest that mitochondrial autophagy-related pathways may contribute to the pathogenesis and progression of UC through metabolic remodeling, immune regulation, and cell type-specific responses. Although these findings require further validation, this study provides novel insights into understanding the molecular mechanisms of UC and facilitates the screening of potential diagnostic biomarkers and therapeutic targets.

## Supporting information

**S1 Fig. Overall design and flowchart.**
(PDF)

**S2 Fig. PPI interaction network.**
(PDF)

**S3 Fig. miRNA of hub genes, TF-prediction network.**
(PDF)

**S4 Fig. Original protein images.**
(PDF)

**S1 Table. Information of datasets.**
(PDF)

**S2 Table. List of gene symbol of MRGs.**
(PDF)

**S3 Table. The primer information.**
(PDF)

**S4 Table. List of gene symbol of MRDEGs.**
(PDF)

**S5 Table. mRNA-miRNA interactions.**
(PDF)

**S6 Table. mRNA-TF interactions.**
(PDF)

**S7 Table. Univariate and Multivariate logistic regression.**
(PDF)

**S1 File. Raw-images.** Original protein images and protein statistical chart.
(PDF)

## Acknowledgments

This work has greatly benefited from the contributions of GEO. We express our gratitude to the GEO network for generously sharing substantial amounts of data, and thank the Yunnan Provincial Engineering Research Central of Preventative Treatment of Traditional Chinese Medicine, In addition, the authors would like to thank the editors and the anonymous reviewers for their valuable comments and suggestions to improve the quality of the paper.

## Author contributions

**Conceptualization:** Yan Qi.

**Data curation:** Junli Shao, Jing Xu.

**Formal analysis:** Jianguo Ma, Rongyi Xu, Yunfei Liu.

**Funding acquisition:** Yan Qi.

**Methodology:** Yunfei Liu.

**Project administration:** Yan Qi.

**Software:** Jianguo Ma, Junli Shao.

**Supervision:** Yan Qi.

**Validation:** Rongyi Xu, Yunfei Liu, Jing Xu.

**Visualization:** Junli Shao.

**Writing – original draft:** Jianguo Ma, Rongyi Xu.

**Writing – review & editing:** Yan Qi.

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
