## [Decision Letter · Decision Letter 0]

28 Sep 2025

Integrated RNA-seq and scRNA-seq to explore the biological mechanisms of mitophagy-related genes in ulcerative colitis

PLOS ONE

Dear Dr. Qi,

Thank you for submitting your manuscript to PLOS ONE. After careful consideration, we feel that it has merit but does not fully meet PLOS ONE’s publication criteria as it currently stands. Therefore, we invite you to submit a revised version of the manuscript that addresses the points raised during the review process.

We look forward to receiving your revised manuscript.

Kind regards,

Alexis G. Murillo Carrasco

Academic Editor

PLOS ONE

Journal Requirements:

“The National Natural Science Foundation of China (Grant number [81960868]) and the Yunnan Provincial Science and Technology Department, funded by the Applied Basic Research Joint SpecialFunds of Yunnan University of Chinese Medicine (Grant numbers [202001AZ070001-051] and [202101AZ070001-013]), High-level TCM talents: (reserve talents) project of Yunnan (2021. No. 1), Xingdian Talent Support Program-Youth Talent Special Project (2023. No. 166), Yunnan Provincial Science and Technology Department - Yunnan University of Traditional Chinese Medicine Joint special Project (Grant numbers [202201AG070192]), Applied Basic Research Key Project of Yunnan (202501AS070155), Yunnan Applied Basic Research Project--Joint Project of Traditional Chinese Medicine and Surface(202001AZ070001-051).”

Please state what role the funders took in the study.  If the funders had no role, please state: 'The funders had no role in study design, data collection and analysis, decision to publish, or preparation of the manuscript.'

Additional Editor Comments:

Please revise all reviewers' comments and respond to them accordingly. In addition, please be sure to describe if you applied p-adjustment for multiple comparisons (for example, for generating the volcano plot). If so, please include the strategy applied (BH, FDR, Others).

According to the figure shown in the Nomogram, there is no clear reference to the risk related to these markers. Perhaps it is relevant to test the MRG-based score in combination with other clinical profiles instead of combining all relevant genes? How can it be applied to the routine? In the current Figure 6D, it is hard to see the contribution of each gene, particularly in the cases of PPARGC1A and MIF. Please revise it.

Please describe in high-level details the methods followed to annotate potential cell types in the single-cell dataset. How did you define which cell types could be present in your sample? Following the previous comment, I suggest showing a dotplot with relevant genes from Figure 9, organized by cell type and donor type (UC or control).

Finally, please add statistical comparisons in Figure 10C and attach the melting curves for SYBR-amplified regions.

Reviewers' comments:

Reviewer's Responses to Questions

**Comments to the Author**

1. Is the manuscript technically sound, and do the data support the conclusions?

Reviewer #1: Yes

Reviewer #2: Yes

Reviewer #3: Yes

2. Has the statistical analysis been performed appropriately and rigorously?

Reviewer #1: Yes

Reviewer #2: Yes

Reviewer #3: No

3. Have the authors made all data underlying the findings in their manuscript fully available?

Reviewer #1: Yes

Reviewer #2: Yes

Reviewer #3: Yes

4. Is the manuscript presented in an intelligible fashion and written in standard English?

Reviewer #1: Yes

Reviewer #2: Yes

Reviewer #3: Yes

Reviewer #1: In this manuscript, Ma et al. identified 13 key genes involved in ulcerative colitis through bioinformatic analysis of publicly available next-generation DNA sequencing raw data. Furthermore, the authors validated their observations on a bench using a mouse model of ulcerative colitis. The strengths of the study design are as follows: 1) Gene prioritization was performed using RNA-seq data from human tissues, narrowing down to hub and key genes; 2) single-cell RNA-seq data were used; and 3) expression changes in mice were confirmed by RT-qPCR and Western blotting. However, mechanistic verification is lacking in research design. In other words, experiments such as using the identified gene knock-out mice, knock-down experiment using siRNA, or experiments using inhibitors of the identified protein. However, the reviewer has never requested additional experiments for any review. Such so-called "Reviewer Experiments" are, in fact, not very meaningful in academic papers (Nature 2011;472:391). The authors should construct their logic and draw conclusions based solely on the data presented. Furthermore, most PLOS ONE values represent the validity and robustness of the methodology. Therefore, reviewers should point out the following aspects concerning the validity of the methods rather than the significance of the author's research results.

Overall, the proposed methodology is reasonable. However, as the authors themselves indicate in the title, the most controversial point is the integration of RNA-seq and scRNA-seq results as well as subsequent gene prioritization. There is still no established consensus on this methodology, with various reports and debates. The authors should discuss this issue in the Discussion section.

Reviewer #2: Ma et al aims to decipher the mechanism of mitophagy related genes in ulcerative colitis. Overall, it is a well designed, executed and validated work. Multi-modal tools were utilized to expand spectrum of evidence representation.

Here are some points to note:

1. It is understandable that the diagnostic model developed by the authors is still in the early stage, given that more datasets have to be used for cross-validation. Could the authors deliberate more of the potential use, as well as the sensitivity and specificity of limitation of detection?

2. Following point 1, what kind of human samples to be needed for such a test? Turnaround duration from sample to data, and to diagnostic result? Estimated cost per test? Benefit population figures from such a test of UC patients?

Reviewer #3: Dear Authors,

I think that this manuscript could benefit from a substantial revision - because of the points below, the main flow of the argument is made difficult to follow.

Main points:

1. References in the text to figures and tables are sometimes incorrect and confused, making the flow of the text difficult to follow. In particular:

- Line 248 - Supplementary Figure S1 is not a flowchart of the study, but bar plots of gene ontology analyses

- Line 257 states 21665 DEGs, but there are only 1802 shown in figure 2B (this might be a misreading on my part) - and also the reference to the comparison between MRGs and DEGs should refer to figure 2B rather than 2A

- Line 311-317 - the genes in figure S3A are ALDH6A1, ACADSB, ALDH2, PC, ALDH1A1, and ACSL1 but the text around these lines does not refer to these, but to other datasets

- Lines 325-335 refer to cluster analysis for the genes, but this is not labelled on the plot in 4B

- Line 343/figure 4G - the figure contains genes not written in the text here.

- Line 317 - Table S5 in the text is actually Table S6

- Line 323 - Table S6 in the text is actually Table S7

- Line 391 - Table S7 in the text is actually Table S8

- Line 408-409 mentioned 15000 variable genes in the scRNA analysis, but the figure 7B shows 2000 variable genes

- Lines 429-430 have the labels for figures 8B and 8C swapped in the text

- Also Figure 8D is referred to as a heat map, but it is a plot of the expression of seven chosen genes shown on the UMAP

2. Supplementary Figure S1 and Supplementary Table S5 both report gene ontology analysis results but are not referred to in the text, in the methods, results or the discussion. If this analysis was performed on this data and is to be presented in the manuscript, then it should be reported on and discussed in the context of MRGs and/or ulcerative colitis.

3. Figure 6D shows the nomogram for the predictor - I'm not certain what the point of this diagram is in the manuscript. What if the ranges of expression of these genes are outside the values on the scales? Do we take the minimum or the maximum of the point score? How does the linear predictor and the risk get calculated from the values here?

4. The main figures are low resolution (as they currently stand, maybe a high resolution version is available?). This particularly affects figure 9A and 9B where the gene labels are difficult to read because of resolution and the compression artefacts.

5. The materials and methods appear to consistently refer to raw p-values. It it clear from some of the later text in the results that some of these have been adjusted for multiple comparisons - this is not mentioned in the methods section and should be made clear here.

Some more specific points to make:

1. Figure 1C - PC2 is shown as explaining 'a5.8%' of the variance.

2. In figure 2C, what is the score that is plotted in the heatmap?

3. Figure 2D has the word 'adataset' (possible typo) in the title. There is also no label on the red-white-blue scale

4. Figure 5B/line 358-359: place the labels on figure 5B in the order they are in the text, or write the text in the same order as the labels on Figure 5B

5. Figure 6C - this would be better shown as log-HRs on the x axis because of the scales on the x axis. This would make the difference between the calculated CIs and the HR 0 line clearer to see

6. Figure 10A - label on the colon images which is the diseased and which is the normal.

7. Figure 11 - order the datastet by placing the significant genes first: also place the gene name as a title on each pair of bars for the bar plot

8. Figure 12A - What are the two bands that appear on the blot for the MIF protein? Which one has been taken as the one to quantify?

9. Figure 12B - this would benefit from the individual blot quantifications in the two conditions shown on each bar - the blot images show significant variation (eg NAMPT in the NC condition, and the top MIF band in the UC condition). It is also not mentioned what the intensity was quantified relative to.

.

Reviewer #1: **Yes:** Shigekazu SuginoShigekazu SuginoShigekazu SuginoShigekazu Sugino

Reviewer #2: No

Reviewer #3: No

---

## [Author Response · Author response to Decision Letter 1]

11 Nov 2025

Response Letter

Your comments, along with those of the other reviewers, are extremely important to our article. We greatly appreciate your insights, which have significantly contributed to enhancing the quality of our work. In this revision, we have prepared a detailed point-by-point response letter for you and the other valued reviewers. This letter acknowledges your feedback and clearly outlines the revisions we have made.

Date: Sep 28 2025 11:40PM

To: "Yan Qi" qiyankm@163.com

From: "PLOS ONE" plosone@plos.org

Subject: PLOS ONE Decision: Revision required [PONE-D-25-44869]

PONE-D-25-44869

Integrated RNA-seq and scRNA-seq to explore the biological mechanisms of mitophagy-related genes in ulcerative colitis

PLOS ONE

Dear Dr. Qi,

Thank you for submitting your manuscript to PLOS ONE. After careful consideration, we feel that it has merit but does not fully meet PLOS ONE’s publication criteria as it currently stands. Therefore, we invite you to submit a revised version of the manuscript that addresses the points raised during the review process.

A rebuttal letter that responds to each point raised by the academic editor and reviewer(s). You should upload this letter as a separate file labeled 'Response to Reviewers'.

A marked-up copy of your manuscript that highlights changes made to the original version. You should upload this as a separate file labeled 'Revised Manuscript with Track Changes'.

An unmarked version of your revised paper without tracked changes. You should upload this as a separate file labeled 'Manuscript'.

If applicable, we recommend that you deposit your laboratory protocols in protocols.io to enhance the reproducibility of your results. Protocols.io assigns your protocol its own identifier (DOI) so that it can be cited independently in the future. For instructions see:https://journals.plos.org/plosone/s/submission-guidelines#loc-laboratory-protocols. Additionally, PLOS ONE offers an option for publishing peer-reviewed Lab Protocol articles, which describe protocols hosted on protocols.io. Read more information on sharing protocols at https://plos.org/protocols?utm_medium=editorial-email&utm_source=authorletters&utm_campaign=protocols.

We look forward to receiving your revised manuscript.

Kind regards,

Alexis G. Murillo Carrasco

Academic Editor

PLOS ONE

Journal Requirements:

Comment 1: Thank you for your careful review and valuable comments.We will rigorously verify all submitted documents against PLOS ONE's formatting guidelines and templates, ensuring proper naming conventions and content formatting to guarantee full compliance with the journal's latest requirements. Thank you again for your valuable suggestion.

“The National Natural Science Foundation of China (Grant number [81960868]) and the Yunnan Provincial Science and Technology Department, funded by the Applied Basic Research Joint SpecialFunds of Yunnan University of Chinese Medicine (Grant numbers [202001AZ070001-051] and [202101AZ070001-013]), High-level TCM talents: (reserve talents) project of Yunnan (2021. No. 1), Xingdian Talent Support Program-Youth Talent Special Project (2023. No. 166), Yunnan Provincial Science and Technology Department - Yunnan University of Traditional Chinese Medicine Joint special Project (Grant numbers [202201AG070192]), Applied Basic Research Key Project of Yunnan (202501AS070155), Yunnan Applied Basic Research Project--Joint Project of Traditional Chinese Medicine and Surface(202001AZ070001-051)”.

Please state what role the funders took in the study. If the funders had no role, please state: 'The funders had no role in study design, data collection and analysis, decision to publish, or preparation of the manuscript.'

Comment Financial Disclosure:

Thank you for your careful review and valuable comments. I have determined that all of the items mentioned in the financial disclosure provide financial support for this article. The funders had no role in study design, data collection and analysis, decision to publish, or preparation of the manuscript. Thank you again for your valuable suggestion.

3.When completing the data availability statement of the submission form, you indicated that you will make your data available on acceptance. We strongly recommend all authors decide on a data sharing plan before acceptance, as the process can be lengthy and hold up publication timelines. Please note that, though access restrictions are acceptable now, your entire data will need to be made freely accessible if your manuscript is accepted for publication. This policy applies to all data except where public deposition would breach compliance with the protocol approved by your research ethics board. If you are unable to adhere to our open data policy, please kindly revise your statement to explain your reasoning and we will seek the editor's input on an exemption. Please be assured that, once you have provided your new statement, the assessment of your exemption will not hold up the peer review process.

Comment: Thank you for your careful review of my manuscript.I confirm that the data will be made available after the paper is accepted.

4.PLOS ONE now requires that authors provide the original uncropped and unadjusted images underlying all blot or gel results reported in a submission’s figures or Supporting Information files. This policy and the journal’s other requirements for blot/gel reporting and figure preparation are described in detail at https://journals.plos.org/plosone/s/figures#loc-blot-and-gel-reporting-requirements and https://journals.plos.org/plosone/s/figures#loc-preparing-figures-from-image-files. When you submit your revised manuscript, please ensure that your figures adhere fully to these guidelines and provide the original underlying images for all blot or gel data reported in your submission. See the following link for instructions on providing the original image data: https://journals.plos.org/plosone/s/figures#loc-original-images-for-blots-and-gels.

Comment 4: Thank you for your careful review of my manuscript. I have read the relevant requirements of this journal for protein immunoassay, and I have uploaded the original image of protein immunoassay as a Supporting Information according to the submission requirements, In the cover letter, blot image data has been clearly written as supporting information to be uploaded.

5.If the reviewer comments include a recommendation to cite specific previously published works, please review and evaluate these publications to determine whether they are relevant and should be cited. There is no requirement to cite these works unless the editor has indicated otherwise.

Comment 5: Thank you for your careful review of my manuscript. I have carefully reviewed this manuscript, and the reviewer did not raise any questions about the references. Thank you again for your valuable suggestions.

Additional Editor Comments:

Please revise all reviewers' comments and respond to them accordingly. In addition, please be sure to describe if you applied p-adjustment for multiple comparisons (for example, for generating the volcano plot). If so, please include the strategy applied (BH, FDR, Others).

According to the figure shown in the Nomogram, there is no clear reference to the risk related to these markers. Perhaps it is relevant to test the MRG-based score in combination with other clinical profiles instead of combining all relevant genes? How can it be applied to the routine? In the current Figure 6D, it is hard to see the contribution of each gene, particularly in the cases of PPARGC1A and MIF. Please revise it.

Please describe in high-level details the methods followed to annotate potential cell types in the single-cell dataset. How did you define which cell types could be present in your sample? Following the previous comment, I suggest showing a dotplot with relevant genes from Figure 9, organized by cell type and donor type (UC or control).

Finally, please add statistical comparisons in Figure 10C and attach the melting curves for SYBR-amplified regions.

Reviewers' comments:

Reviewer's Responses to Questions

Comments to the Author

1. Is the manuscript technically sound, and do the data support the conclusions?

Reviewer #1: Yes

Reviewer #2: Yes

Reviewer #3: Yes

2. Has the statistical analysis been performed appropriately and rigorously?

Reviewer #1: Yes

Reviewer #2: Yes

Reviewer #3: No

3. Have the authors made all data underlying the findings in their manuscript fully available?

ThePLOS Data policyrequires authors to make all data underlying the findings described in their manuscript fully available without restriction, with rare exception (please refer to the Data Availability Statement in the manuscript PDF file). The data should be provided as part of the manuscript or its supporting information, or deposited to a public repository. For example, in addition to summary statistics, the data points behind means, medians and variance measures should be available. If there are restrictions on publicly sharing data—e.g. participant privacy or use of data from a third party—those must be specified.

Reviewer #1: Yes

Reviewer #2: Yes

Reviewer #3: Yes

4. Is the manuscript presented in an intelligible fashion and written in standard English?

Reviewer #1: Yes

Reviewer #2: Yes

Reviewer #3: Yes

5. Review Comments to the Author

Reviewer #1: In this manuscript, Ma et al. identified 13 key genes involved in ulcerative colitis through bioinformatic analysis of publicly available next-generation DNA sequencing raw data. Furthermore, the authors validated their observations on a bench using a mouse model of ulcerative colitis. The strengths of the study design are as follows: 1) Gene prioritization was performed using RNA-seq data from human tissues, narrowing down to hub and key genes; 2) single-cell RNA-seq data were used; and 3) expression changes in mice were confirmed by RT-qPCR and Western blotting. However, mechanistic verification is lacking in research design. In other words, experiments such as using the identified gene knock-out mice, knock-down experiment using siRNA, or experiments using inhibitors of the identified protein. However, the reviewer has never requested additional experiments for any review. Such so-called "Reviewer Experiments" are, in fact, not very meaningful in academic papers (Nature 2011;472:391). The authors should construct their logic and draw conclusions based solely on the data presented. Furthermore, most PLOS ONE values represent the validity and robustness of the methodology. Therefore, reviewers should point out the following aspects concerning the validity of the methods rather than the significance of the author's research results.

Overall, the proposed methodology is reasonable. However, as the authors themselves indicate in the title, the most controversial point is the integration of RNA-seq and scRNA-seq results as well as subsequent gene prioritization. There is still no established consensus on this methodology, with various reports and debates. The authors should discuss this issue in the Discussion section.

Comment: Thank you for your careful review and valuable comments. Regarding the controversial issues you raised, there is currently no unified standard in this field, and related analytical methods are continuously evolving and improving. In this study, we first systematically screened differentially expressed genes, co-expression modules, and key genes at the bulk RNA-seq level, then combined single-cell transcriptomic data for cell-type-specific localization and functional annotation. This strategy, adopted in multiple high-level studies, can reflect cellular heterogeneity while maintaining overall tissue expression characteristics. We will supplement the analysis of methodological controversies in the discussion section of the revised manuscript, citing recent research advancements to further illustrate the rationale and future directions of this workflow. Specific modifications are detailed in the Discussion section on page 27, line 617-633. Thank you again for your valuable suggestion.

Reviewer #2: Ma et al aims to decipher the mechanism of mitophagy related genes in ulcerative colitis. Overall, it is a well designed, executed and validated work. Multi-modal tools were utilized to expand spectrum of evidence representation.

Here are some points to note:

1. It is understandable that the diagnostic model developed by the authors is still in the early stage, given that more datasets have to be used for cross-validation. Could the authors deliberate more of the potential use, as well as the sensitivity and specificity of limitation of detection?

Comment: Thank you for your careful review and valuable comments. The multi-gene diagnostic model developed in this study demonstrated high accuracy in distinguishing ulcerative colitis (UC) from healthy controls, with an AUC of 0.974, indicating strong diagnostic potential. This model is expected to serve as a molecular tool for early clinical diagnosis, identification of complex cases, and personalized risk assessment, thereby enhancing the precision of UC diagnosis. Current evaluations of model sensitivity and specificity primarily rely on internal validation within public databases. However, detection outcomes are influenced by factors such as sample sources and sequencing platforms, necessitating additional independent cohort data and multi-center studies to further validate their generalization capabilities and clinical diagnostic efficacy. We plan to expand sample types and testing methods in subsequent research, while refining model performance through multi-

---

## [Decision Letter · Decision Letter 1]

17 Dec 2025

Dear Dr. Qi,

Thank you for submitting your manuscript to PLOS ONE. After careful consideration, we feel that it has merit but does not fully meet PLOS ONE’s publication criteria as it currently stands. Therefore, we invite you to submit a revised version of the manuscript that addresses the points raised during the review process.

We look forward to receiving your revised manuscript.

Kind regards,

Alexis G. Murillo Carrasco

Academic Editor

PLOS One

Journal Requirements:

Reviewers' comments:

Reviewer's Responses to Questions

**Comments to the Author**

Reviewer #1: All comments have been addressed

Reviewer #2: All comments have been addressed

Reviewer #4: (No Response)

Reviewer #5: (No Response)

2. Is the manuscript technically sound, and do the data support the conclusions?

Reviewer #1: (No Response)

Reviewer #2: Yes

Reviewer #4: Yes

Reviewer #5: No

3. Has the statistical analysis been performed appropriately and rigorously?

Reviewer #1: (No Response)

Reviewer #2: Yes

Reviewer #4: Yes

Reviewer #5: No

4. Have the authors made all data underlying the findings in their manuscript fully available?

Reviewer #1: (No Response)

Reviewer #2: Yes

Reviewer #4: Yes

Reviewer #5: No

5. Is the manuscript presented in an intelligible fashion and written in standard English?

Reviewer #1: (No Response)

Reviewer #2: Yes

Reviewer #4: Yes

Reviewer #5: No

Reviewer #1: Well done! I (reviewer 1) have checked your improvement in the revised manuscript. In line 631, I have found a typo: a double period.

Reviewer #2: Please consider including authors' response to my point 2 comment into the Discussions part. Thanks.

Reviewer #4: The authors first screened differentially expressed genes and key modules via bulk RNA-seq, then employed scRNA-seq for cell type localization and functional annotation. This common “global-to-local” analytical approach facilitates revealing cellular heterogeneity while preserving tissue-wide expression patterns. Similar integration strategies have been applied in high-impact papers in recent years, particularly in disease mechanism studies, providing more refined cell-level insights. Key genes were validated in animal models via qPCR and WB, enhancing the credibility of findings. Clinically, four public transcriptomic datasets and one single-cell dataset were utilized, ensuring substantial sample size, with experimental validation through animal models. The constructed gene diagnostic model achieved an AUC of 0.974, demonstrating strong discriminatory power and potential clinical translational value. As an immune-mediated inflammatory disease, UC exhibits significant correlations between key genes and M1 macrophages, activated T cells, and others identified via CIBERSORT analysis, consistent with its immunopathological characteristics. Some key genes—such as BNIP3, NAMPT, and MIF—have established links to UC pathogenesis through established research in pathways including inflammation, apoptosis, and metabolism.

Main issues:

No standardized workflow currently exists for integrating bulk and scRNA-seq data, with differing methodologies potentially yielding inconsistent results. Bulk data primarily originate from public databases (often tissue samples), while scRNA-seq data may derive from diverse platforms or sample types, introducing batch effects and platform biases. Relying solely on bulk data for gene selection risks overlooking genes highly expressed in specific cell subpopulations but exhibiting weak overall signals. Single-cell annotation relies on known marker genes; incomplete labeling or cell state transitions may introduce annotation errors. These issues warrant close attention from authors to identify and reduce potential biases.

Limitations and discussion:

All data originate from colon tissue, neglecting the clinical potential of non-invasive samples like blood or stool, resulting in limited sample diversity. Additionally, the diagnostic model was validated only within internal data, requiring external cohorts or multicenter data to further validate its generalizability. While key genes were found to correlate with immune cell infiltration, subsequent functional experiments did not elucidate specific regulatory mechanisms. The dataset lacks detailed information on clinical staging, medication history, and disease activity, limiting the model's clinical stratification capability.

Reviewer #5: Thank you for the opportunity to review your manuscript. While I see this is a second submission (a revision), I think there are few overarching areas that still require attention.

The first is that the introduction needs some re-framing and could be more streamlined. Most notably is that mitophagy needs to be clearly defined. The role of mitophagy in IBD (or what is known to date) also needs to be more explicit (e.g. lines 54-61 seem contradictory to lines 61-63, and lines 63-65 is vague and requires a reference).

Second, I have a few concerns/questions regarding the methodological approach and related results:

1) The mitophagy related genes (MRGs), were empirically picked from an online database (GeneCards), and the ones that overlapped with your DEGs were picked for further analysis – but overall there was no evidence to support these 722 genes over any other grouping of genes. An enrichment (hypergeometric) test (from 21,000 genes with 1,800 passing p<0.05, and then looking for overlap with the 722 empirically derived list) indicates that one would expect ~>=61 genes overlapping by chance alone. This exceeds the n=50 gene overlap. Furthermore, it looks as though it was an approx. 50/50 split between those 50 MRGs with increased expression and decreased expression in UC vs control. Overall, this leads me to believe that the overall expression pattern or biology of these transcripts is not that relevant to the GEO samples used in this study. Can the authors please explain why the above 50 MRGs might still be of biological relevance in IBD? How many of the 50 MRGs had an FDR>0.05 from the initial comparison (UC vs control) above?

2) This is an inherent issue in using retrospective, publicly available databases (like GEO). But no information is provided on the clinical status/phenotypes of the patients/samples used, e.g. remission vs. active disease, Mayo or Montreal scores (or otherwise). Thus it is impossible to relate any of the molecular findings back to the clinical phenotype. Subsequently, related to the model that was developed, I think any statement claiming “diagnostic” capabilities is an overstatement, thus this needs to be re-worded. Importantly, this model was not tested against relevant differentials (e.g. Crohn’s colitis). The AUC of 0.97 is concerning for overfitting, and a more likely explanation is that the model is measuring inflammation (this makes sense considering the correlation with inflammatory markers/cells), rather any independent process related to mitophagy or mitophagy as causative factor in IBD.

3) the paper discusses integrating bulk RNA and single cell RNA seq, but some of the GEO data was Affymetrix microarray data, was it not? Please clarify the input technologies and any relevant clinical data (see comments above), and adjust the text as appreciate.

4) PCR, Western blot, or other assays are not validation. Validation would be taking an independent cohort (a test set) and seeing how the MRGs perform in that cohort, using the same technology (e.g. Affy microarrays, RNA seq etc) as before. DSS colitis is not UC, differences are expected, but should be explored/explained. I actually think the in vivo model, PCR and Western blot sections weaken the paper, and more emphasis should be made on refining the other sections. But I will leave this up to the authors and AE/EI.

5) the scRNA seq data is confusing to me. Genes that differentiate cell types from one another are highlighted, but it seems like immunoglobulin genes are highlighted in T cells and other non-lymphocytes. I assume this is because these genes are so highly expressed in B cells, and nonexistent in the other cells. This requires further clarification though why you are highlighting this – the lack of immunoglobulin doesn’t define any particular cell type other than tell us we are looking at something that is probably not from a B cell lineage. (but the direction of these genes/transcripts is not clear based on the text.)

6) Line 89 – what was the inclusion/exclusion criteria? What/how were samples deemed irrelevant?

7) Line 101 – how was epigenetics involved in the GeneCards search? What type of epigenetics?

Last, the paper needs to be checked for minor typos, e.g. line 22 – should be a period, not a comma; Line 238 – space missing between “andP”; Line 257 – space missing between “inbatch”.

---

## [Author Response · Author response to Decision Letter 2]

10 Mar 2026

We would like to express our sincere gratitude to you and the reviewers for your valuable comments on our manuscript. In light of the thorough and insightful feedback provided, we have revised the manuscript accordingly. We have added relevant content to address the reviewers' suggestions, and we have rewritten several sections to ensure clarity and completeness. We hope that these modifications meet the reviewers' expectations and enhance the overall quality of the manuscript.

Reviewer #1: Well done! I (reviewer 1) have checked your improvement in the revised manuscript. In line 631, I have found a typo: a double period.

Comment: Thank you for your thorough review and valuable feedback. We appreciate your recognition of our work. In line with your suggestions, we will standardize all revisions to use English periods to ensure consistency with language and formatting standards. For specific modifications, please refer to page 27, line 631. We sincerely appreciate your valuable input.

Reviewer #2: Please consider including authors' response to my point 2 comment into the Discussions part. Thanks.

Comment: Thank you for your detailed review and valuable feedback. We have refined and consolidated the previous round of explanations regarding the clinical applicability of the diagnostic model as per your comment No.2, and have now incorporated them into the relevant paragraphs of the revised discussion section to systematically present the practical application prospects and potential clinical value of the model in the main text. For specific modifications, please refer to page30-31, lines 718-732. We sincerely appreciate your valuable suggestions once again.

Reviewer #4: The authors first screened differentially expressed genes and key modules via bulk RNA-seq, then employed scRNA-seq for cell type localization and functional annotation. This common “global-to-local” analytical approach facilitates revealing cellular heterogeneity while preserving tissue-wide expression patterns. Similar integration strategies have been applied in high-impact papers in recent years, particularly in disease mechanism studies, providing more refined cell-level insights. Key genes were validated in animal models via qPCR and WB, enhancing the credibility of findings. Clinically, four public transcriptomic datasets and one single-cell dataset were utilized, ensuring substantial sample size, with experimental validation through animal models. The constructed gene diagnostic model achieved an AUC of 0.974, demonstrating strong discriminatory power and potential clinical translational value. As an immune-mediated inflammatory disease, UC exhibits significant correlations between key genes and M1 macrophages, activated T cells, and others identified via CIBERSORT analysis, consistent with its immunopathological characteristics. Some key genes—such as BNIP3, NAMPT, and MIF—have established links to UC pathogenesis through established research in pathways including inflammation, apoptosis, and metabolism.

Main issues:

No standardized workflow currently exists for integrating bulk and scRNA-seq data, with differing methodologies potentially yielding inconsistent results. Bulk data primarily originate from public databases (often tissue samples), while scRNA-seq data may derive from diverse platforms or sample types, introducing batch effects and platform biases. Relying solely on bulk data for gene selection risks overlooking genes highly expressed in specific cell subpopulations but exhibiting weak overall signals. Single-cell annotation relies on known marker genes; incomplete labeling or cell state transitions may introduce annotation errors. These issues warrant close attention from authors to identify and reduce potential biases.

Comment: We appreciate the reviewers' insightful suggestions regarding the integration of bulk and scRNA-seq analytical methodologies. We fully agree that there is currently no universally accepted standard workflow for bulk-scRNA integration, and different strategies may indeed introduce biases. Therefore, in this study, we adopted a macro-to-micro approach: "first conducting robust candidate gene screening using tissue-level bulk data, followed by cell type localization and refined interpretation of expression patterns using single-cell data," to maximize the advantages of sample size while maintaining cellular resolution. Specifically, at the bulk level, we integrated multiple independent cohorts, performed rigorous batch effect correction (sva/Combat), and conducted differential analysis combined with WGCNA screening, while cross-referencing with an independent mitochondrial autophagy gene set to identify candidate genes. This approach enhances signal stability across larger sample sizes and reduces the impact of single-cohort or platform biases. At the scRNA-seq level, we utilized Harmony for cross-sample integration, performed cell annotation based on SingleR and established marker profiles from the literature, and focused primarily on the relative enrichment and localization of the aforementioned candidate genes within major cell populations, rather than "discovering new genes" from single-cell data alone, to minimize biases caused by platform differences and annotation uncertainties. We acknowledge the two key limitations identified by the reviewers: first, the reliance on bulk data for initial screening may miss signals that are highly expressed only in small cell subsets; second, single-cell annotation itself is subject to marker dependency and errors caused by state transitions. Under the current conditions, the contribution of this study lies primarily in refining the identification of mitochondrial autophagy-related candidate genes from the "tissue-wide level" to specific cell types through this integrated framework, thereby providing more focused targets for subsequent targeted functional experiments and multi-cohort validation. We will add a dedicated section in the revised Discussion to systematically elaborate on the rationale, advantages, and limitations of our bulk–scRNA integration strategy, and explicitly address the aforementioned potential biases and future improvement directions (e.g., incorporating multi-cohort single-cell data or adopting more refined subset resolution methods) to enhance readers' comprehensive understanding of the methodological positioning of this study. Specific revisions are detailed on Page 27, Lines 634–646. We once again appreciate your valuable feedback.

Limitations and discussion:

All data originate from colon tissue, neglecting the clinical potential of non-invasive samples like blood or stool, resulting in limited sample diversity. Additionally, the diagnostic model was validated only within internal data, requiring external cohorts or multicenter data to further validate its generalizability. While key genes were found to correlate with immune cell infiltration, subsequent functional experiments did not elucidate specific regulatory mechanisms. The dataset lacks detailed information on clinical staging, medication history, and disease activity, limiting the model's clinical stratification capability.

Comment: We appreciate the reviewers 'summary and analysis of the study's limitations. We acknowledge that all current data are derived from colon tissue and do not yet cover non-invasive samples with higher clinical accessibility, such as blood and stool, which indeed limits the breadth of sample types and clinical application scenarios. The gene combination constructed in this study has only been evaluated in the merged internal cohort and lacks validation in independent external or multicenter cohorts. We agree that it is currently more appropriate to classify it as an 'exploratory classification/auxiliary identification model' rather than a mature clinical diagnostic tool. Additionally, although we found that key genes are significantly associated with the degree of infiltration by various immune cells and provided preliminary biological support at the mRNA and protein levels in the DSS colitis model, we have not yet systematically elucidated the specific mechanisms by which these genes regulate immune responses and mitochondrial autophagy within specific cell subpopulations through in-depth functional experiments. Furthermore, due to the lack of detailed clinical information such as disease activity, medication history, and staging/typing in the open cohort, it is currently difficult to further evaluate the potential application of these molecular features in clinical stratification and prognostic prediction. We will add an independent section in the Discussion of the revised manuscript to systematically present the aforementioned limitations and explicitly state that the primary contribution of this study lies in integrating tissue transcriptome and single-cell data, proposing a set of candidate genes and a preliminary classification model associated with mitochondrial autophagy and immune infiltration. Future validation will still require external confirmation based on prospective, multicenter, and multi-sample population cohorts, combined with in-depth functional experiments to elucidate their specific regulatory mechanisms and clinical application value. Specific revisions are detailed on pages 27-28, lines 647–670. We sincerely appreciate your valuable feedback.

Reviewer #5: Thank you for the opportunity to review your manuscript. While I see this is a second submission (a revision), I think there are few overarching areas that still require attention.

The first is that the introduction needs some re-framing and could be more streamlined. Most notably is that mitophagy needs to be clearly defined. The role of mitophagy in IBD (or what is known to date) also needs to be more explicit (e.g. lines 54-61 seem contradictory to lines 61-63, and lines 63-65 is vague and requires a reference).

Comment: Thank you for your feedback. Based on your suggestions, we will further clarify the definition of mitochondrial autophagy in the revised draft, summarize its existing research progress in IBD, and adjust relevant expressions to avoid contradictions and ambiguities. Specific modifications are detailed on Pages 3-4, Lines 54-71. We appreciate your valuable input once again.

Second, I have a few concerns/questions regarding the methodological approach and related results:

1) The mitophagy related genes (MRGs), were empirically picked from an online database (GeneCards), and the ones that overlapped with your DEGs were picked for further analysis – but overall there was no evidence to support these 722 genes over any other grouping of genes. An enrichment (hypergeometric) test (from 21,000 genes with 1,800 passing p<0.05, and then looking for overlap with the 722 empirically derived list) indicates that one would expect ~>=61 genes overlapping by chance alone. This exceeds the n=50 gene overlap. Furthermore, it looks as though it was an approx. 50/50 split between those 50 MRGs with increased expression and decreased expression in UC vs control. Overall, this leads me to believe that the overall expression pattern or biology of these transcripts is not that relevant to the GEO samples used in this study. Can the authors please explain why the above 50 MRGs might still be of biological relevance in IBD? How many of the 50 MRGs had an FDR>0.05 from the initial comparison (UC vs control) above?

Comment: We appreciate the reviewers' thorough consideration of the rationale for MRG selection and statistical validity. Regarding the source of the gene set, we employed genes with high functional annotation relevance to mitophagy from GeneCards and MSigDB as the "mitochondrial autophagy candidate background set." The objective was not to demonstrate overall significant enrichment of the entire 772-gene set in UC, but rather to further narrow down a subset of candidate genes that are indeed expressed with altered expression patterns in UC colonic tissues and located in modules highly correlated with the phenotype, by integrating differential analysis and WGCNA network screening from established mitochondrial/autophagy-related pathways. Consequently, we focused more on "key node genes screened under the mitophagy-related context" rather than whether the overlap quantity exceeded random expectations or exhibited a single up/downward direction. In fact, many of these genes (e.g., BNIP3, NAMPT, MIF, etc.) have functional evidence from previous IBD or inflammation/metabolism studies. By integrating multi-cohort differential analysis, co-expression networks, and immune infiltration associations, this study further localized these molecules from large-scale annotation sets to specific UC-related expression patterns and cell type localization, thereby providing clearer clues for functional hypotheses and subsequent mechanistic research directions. Regarding the issue of p-values and FDR, we have adopted a unified statistical threshold framework in the differential analysis and employed multiple correction to control for the overall false-positive risk in the enrichment analysis. Given that the objective of this study is to propose candidate genes and potential mechanistic clues rather than to establish a rigorous "mitophagy enrichment statistical test," we believe the current analytical depth is sufficient to support the research goals. We appreciate your valuable feedback once again.

2) This is an inherent issue in using retrospective, publicly available databases (like GEO). But no information is provided on the clinical status/phenotypes of the patients/samples used, e.g. remission vs. active disease, Mayo or Montreal scores (or otherwise). Thus it is impossible to relate any of the molecular findings back to the clinical phenotype. Subsequently, related to the model that was developed, I think any statement claiming “diagnostic” capabilities is an overstatement, thus this needs to be re-worded. Importantly, this model was not tested against relevant differentials (e.g. Crohn’s colitis). The AUC of 0.97 is concerning for overfitting, and a more likely explanation is that the model is measuring inflammation (this makes sense considering the correlation with inflammatory markers/cells), rather any independent process related to mitophagy or mitophagy as causative factor in IBD.

Comment: Thank you for your constructive feedback. We acknowledge that directly characterizing our model as having 'diagnostic capability' based on retrospective public data such as GEO, in the absence of detailed clinical subtyping information (e.g., activity level, Mayo/Montreal score), may be an overstatement. Moreover, the model has not yet been tested in practical differential diagnostic scenarios such as Crohn's colitis, and the high AUC results are more likely to reflect sensitivity to inflammatory states. In the revised manuscript, we will uniformly soften the related terminology, positioning the model as an exploratory classification/inflammation-associated feature based on mitochondrial autophagy-related genes, rather than a mature diagnostic tool. We will explicitly state in the limitations section of the Discussion that the lack of clinical phenotype information and external disease controls limits the correlation and generalizability of molecular features to specific clinical states. The high AUC should be interpreted with caution, and further validation and comparison in prospective, multicenter cohorts including UC and other intestinal diseases (e.g., Crohn's disease) are required. Specific revisions are detailed on Page 24-25, Line 601-620. Thank you again for your valuable suggestions.

3) the paper discusses integrating bulk RNA and single cell RNA seq, but some of the GEO data was Affymetrix microarray data, was it not? Please clarify the input technologies and any relevant clinical data (see comments above), and adjust the text as appreciate.

Comment: Thank you for your feedback on the data type description. In this s

---

## [Decision Letter · Decision Letter 2]

23 Mar 2026

Dear Dr. Qi,

Thank you for submitting your manuscript to PLOS ONE. After careful consideration, we feel that it has merit but does not fully meet PLOS ONE’s publication criteria as it currently stands. Therefore, we invite you to submit a revised version of the manuscript that addresses the points raised during the review process.

We look forward to receiving your revised manuscript.

Kind regards,

Alexis G. Murillo Carrasco

Academic Editor

PLOS One

Journal Requirements:

Reviewers' comments:

Reviewer's Responses to Questions

**Comments to the Author**

Reviewer #1: All comments have been addressed

Reviewer #2: All comments have been addressed

Reviewer #4: All comments have been addressed

Reviewer #5: All comments have been addressed

2. Is the manuscript technically sound, and do the data support the conclusions?

Reviewer #1: Yes

Reviewer #2: Yes

Reviewer #4: Yes

Reviewer #5: Partly

3. Has the statistical analysis been performed appropriately and rigorously?

Reviewer #1: Yes

Reviewer #2: Yes

Reviewer #4: Yes

Reviewer #5: Yes

4. Have the authors made all data underlying the findings in their manuscript fully available?

Reviewer #1: Yes

Reviewer #2: Yes

Reviewer #4: Yes

Reviewer #5: Yes

5. Is the manuscript presented in an intelligible fashion and written in standard English?

Reviewer #1: Yes

Reviewer #2: Yes

Reviewer #4: Yes

Reviewer #5: Yes

Reviewer #1: The authors improved the sentences the reviewer pointed out. I believe that this research will be published in near future.

Reviewer #2: All good for me. All the comments from me previously have been properly addressed from the authors.

Reviewer #4: The authors have satisfactorily addressed my previous concerns regarding the integration of bulk and single-cell RNA-seq data. The revised manuscript now clearly explains the rationale behind their "global-to-local" analytical strategy and includes a dedicated discussion of its advantages and limitations, which enhances the methodological transparency.

I have only one minor suggestion: there is some repetition in the Discussion section regarding the integration strategy. Streamlining this would improve conciseness.

Overall, the authors have responded adequately to my comments, and the manuscript is now suitable for publication with minor revisions.

Reviewer #5: Thank you for addressing my comments and questions, and making the appropriate changes to the text. The manuscript is now quite long, I estimat approx >11-12,000 words (from Abstract to end of Discussion), and the discussion is approximately 5,000 words itself? (just a rough estimate from looking at the text in word). I think most traditional scientific journals would request that the word count be significantly reduced. But I'll leave this decision up to the journal/Editors.

.

Reviewer #1: **Yes:** Shigekazu SuginoShigekazu SuginoShigekazu SuginoShigekazu Sugino

Reviewer #2: No

Reviewer #4: No

Reviewer #5: No

---

## [Author Response · Author response to Decision Letter 3]

24 Mar 2026

Response Letter

We would like to express our sincere gratitude to you and the reviewers for your valuable comments on our manuscript. In light of the thorough and insightful feedback provided, we have revised the manuscript accordingly. We have added relevant content to address the reviewers' suggestions, and we have rewritten several sections to ensure clarity and completeness. We hope that these modifications meet the reviewers' expectations and enhance the overall quality of the manuscript.

Date: Mar 23 2026 09:54PM

To: "Yan Qi" qiyankm@163.com

From: "PLOS ONE" plosone@plos.org

Subject: PLOS ONE Decision: Revision required [PONE-D-25-44869R2]

PONE-D-25-44869R2

Integrated RNA-seq and scRNA-seq to explore the biological mechanisms of mitophagy-related genes in ulcerative colitis

PLOS One

Dear Dr. Qi,

Thank you for submitting your manuscript to PLOS ONE. After careful consideration, we feel that it has merit but does not fully meet PLOS ONE’s publication criteria as it currently stands. Therefore, we invite you to submit a revised version of the manuscript that addresses the points raised during the review process.

Please revise the discussion section and evaluate implementing the reviewer's comment.

Comment: Thank you for your thorough review and valuable feedback. We will revise the discussion section based on your suggestions, removing redundant content to enhance the article's clarity. Thank you again for your valuable feedback.

A letter that responds to each point raised by the academic editor and reviewer(s). You should upload this letter as a separate file labeled 'Response to Reviewers'.

A marked-up copy of your manuscript that highlights changes made to the original version. You should upload this as a separate file labeled 'Revised Manuscript with Track Changes'.

An unmarked version of your revised paper without tracked changes. You should upload this as a separate file labeled 'Manuscript'.

We look forward to receiving your revised manuscript.

Kind regards,

Alexis G. Murillo Carrasco

Academic Editor

PLOS One

Journal Requirements:

Reviewers' comments:

Reviewer's Responses to Questions

Comments to the Author

1. If the authors have adequately addressed your comments raised in a previous round of review and you feel that this manuscript is now acceptable for publication, you may indicate that here to bypass the “Comments to the Author” section, enter your conflict of interest statement in the “Confidential to Editor” section, and submit your "Accept" recommendation.

Reviewer #1: All comments have been addressed

Reviewer #2: All comments have been addressed

Reviewer #4: All comments have been addressed

Reviewer #5: All comments have been addressed

2. Is the manuscript technically sound, and do the data support the conclusions?

Reviewer #1: Yes

Reviewer #2: Yes

Reviewer #4: Yes

Reviewer #5: Partly

3. Has the statistical analysis been performed appropriately and rigorously?

Reviewer #1: Yes

Reviewer #2: Yes

Reviewer #4: Yes

Reviewer #5: Yes

4. Have the authors made all data underlying the findings in their manuscript fully available?

Reviewer #1: Yes

Reviewer #2: Yes

Reviewer #4: Yes

Reviewer #5: Yes

5. Is the manuscript presented in an intelligible fashion and written in standard English?

Reviewer #1: Yes

Reviewer #2: Yes

Reviewer #4: Yes

Reviewer #5: Yes

6. Review Comments to the Author

Reviewer #1: The authors improved the sentences the reviewer pointed out. I believe that this research will be published in near future.

Comment: Thank you for your careful review and valuable comments. We appreciate your recognition of the manuscript revisions and look forward to the smooth publication of the study. Thank you again for your valuable feedback.

Reviewer #2: All good for me. All the comments from me previously have been properly addressed from the authors.

Comment: Thank you for your review and approval. We are pleased to know that all modifications have met your expectations. Thank you for your recognition of our efforts. Thank you again for your valuable feedback.

Reviewer #4: The authors have satisfactorily addressed my previous concerns regarding the integration of bulk and single-cell RNA-seq data. The revised manuscript now clearly explains the rationale behind their "global-to-local" analytical strategy and includes a dedicated discussion of its advantages and limitations, which enhances the methodological transparency.

I have only one minor suggestion: there is some repetition in the Discussion section regarding the integration strategy. Streamlining this would improve conciseness.

Overall, the authors have responded adequately to my comments, and the manuscript is now suitable for publication with minor revisions.

Comment: Thank you for your careful review and valuable comments. We appreciate your recognition of the manuscript revisions. In response to your suggestions, we will further streamline and organize the redundant descriptions regarding the integration strategy of bulk RNA-seq and single-cell RNA-seq in the Discussion section to enhance the conciseness and readability of the article. Specific modifications are detailed on Page 25-26, Line 602-608. Thank you again for your valuable feedback.

Reviewer #5: Thank you for addressing my comments and questions, and making the appropriate changes to the text. The manuscript is now quite long, I estimat approx >11-12,000 words (from Abstract to end of Discussion), and the discussion is approximately 5,000 words itself? (just a rough estimate from looking at the text in word). I think most traditional scientific journals would request that the word count be significantly reduced. But I'll leave this decision up to the journal/Editors.

Comment: Thank you for your careful review and valuable comments. We appreciate your recognition of the preliminary revisions. Regarding the issue of excessive manuscript length raised by you, we will further streamline and organize the entire text, particularly the Discussion section, while retaining core information and minimizing redundant expressions to enhance the conciseness and readability of the manuscript. Specific revisions are detailed on Page 22-27, Line 522-641, the discussion section totals 1,370 words. Thank you again for your valuable feedback.

7. PLOS authors have the option to publish the peer review history of their article (what does this mean?). If published, this will include your full peer review and any attached files.

Do you want your identity to be public for this peer review? For information about this choice, including consent withdrawal, please see our Privacy Policy.

Reviewer #1: Yes: Shigekazu Sugino

Reviewer #2: No

Reviewer #4: No

Reviewer #5: No

---

## [Editor Report · Decision Letter 3]

27 Mar 2026

Integrated RNA-seq and scRNA-seq to explore the biological mechanisms of mitophagy-related genes in ulcerative colitis

PONE-D-25-44869R3

Dear Dr. Qi,

We’re pleased to inform you that your manuscript has been judged scientifically suitable for publication and will be formally accepted for publication once it meets all outstanding technical requirements.

Kind regards,

Alexis G. Murillo Carrasco

Academic Editor

PLOS One
---

## [Editor Report · Acceptance letter]

PONE-D-25-44869R3

PLOS One

Dear Dr. Qi,

I'm pleased to inform you that your manuscript has been deemed suitable for publication in PLOS One. Congratulations! Your manuscript is now being handed over to our production team.

Kind regards,

on behalf of

Dr. Alexis G. Murillo Carrasco

Academic Editor

PLOS One